# Estimation of the 3-D geoelectric field at the Earth's surface using Spherical Elementary Current Systems

Liisa Juusola[1], Heikki Vanhamäki[2], Elena Marshalko[1], Mikhail Kruglyakov[3], and Ari Viljanen[1]

[1]Finnish Meteorological Institute, Helsinki, Finland
[2]University of Oulu, Oulu, Finland
[3]University of Otago, Dunedin, New Zealand

**Correspondence:** Liisa Juusola (liisa.juusola@fmi.fi)

**Abstract.** The geoelectric field drives geomagnetically induced currents (GIC) in technological conductor networks, which can affect the performance of critical ground infrastructure such as electric power transmission grids. The three-dimensional (3-D) electric field at the Earth's surface consists of an external divergence-free (DF) part due to temporally and spatially varying ionospheric and magnetospheric currents, an internal DF part due to temporally and spatially varying telluric currents, and a curl-free (CF) part due to charge accumulation at ground conductivity gradients. We have developed a new method for estimating these contributions. The external and internal parts of the DF electric field are calculated from the time derivative of the external and internal parts of the observed ground magnetic field, respectively, using DF two-dimensional (2-D) Spherical Elementary Current Systems (SECS). The horizontal surface CF electric field is calculated from the known surface DF electric field using coefficients that linearly relate the DF electric field to the CF electric field. The coefficients were obtained from the 3-D induction model PGIEM2G. The calculations are carried out in the time domain and only two consecutive time steps of the observed magnetic field are needed to compute the surface electric field. The external part of the DF electric field is valid at and below the ionosphere, the internal part at and above the Earth's surface, and the CF part at the Earth's surface. A dense magnetometer network is a requirement for reliable results. The external and internal parts of the DF electric field are generally oppositely directed and have comparable amplitudes, both on the ground and in the ionosphere, indicating that both contributions are significant for the total DF electric field. The largest peaks of total DF electric field tend to occur when either the external or internal contribution is temporarily suppressed. At a given location, a DF electric field with a given amplitude can result in a total surface electric field amplitude with orders of magnitude difference depending on the direction of the DF electric field with respect to the locally dominant conductivity gradient structure. The electric field calculation is computationally light, facilitating operational implementation of a near-real time 3-D surface electric field monitoring and derivation of long electric field time series.

## 1 Introduction

The geoelectric field at the Earth's surface drives geomagnetically induced currents (GIC) in technological conductor networks, which can affect the performance of critical ground infrastructure such as electric power transmission grids (Pulkkinen et al., 2017). In order to avoid incidents caused by space weather, a solid understanding of the possible amplitude range of the highly

variable and location-dependent electric field is needed. Because electric field measurements are scarce, the surface electric field is typically modelled using magnetic field measurements and knowledge of the conductivity of the Earth. For the same reason, validation of the modelling results is difficult, which means that it would be useful to have several alternative approaches that would allow inter-comparison of various modelling results.

Several modelling techniques for estimating the surface electric field exist, either based on one-dimensional (1-D) ground conductivity models (Viljanen et al., 2004, 2012, 2014), magnetotelluric transfer functions (Kelbert et al., 2017; Love et al., 2018; Lucas et al., 2020; Malone-Leigh et al., 2023), or three-dimensional (3-D) ground conductivity models (Rosenqvist and Hall, 2019; Marshalko et al., 2021; Kruglyakov et al., 2022; Marshalko et al., 2023; Kruglyakov et al., 2023). 1-D modelling only considers vertical conductivity gradients, and consequently ignores lateral gradients, which are now known to be highly significant (Marshalko et al., 2021). The magnetotelluric transfer functions take lateral conductivity gradients into account but ignore spatial variations in the source, which are also known to be important (Lucas et al., 2018; Marshalko et al., 2021). 3-D electromagnetic modelling considers both vertical and lateral conductivity gradients and can also take into account spatial variations of the inducing source, but as a result of sparse magnetometer coverage and inaccuracies in 3-D conductivity models, the internal part of the magnetic field variations is not conserved (Kruglyakov et al., 2023; Marshalko et al., 2023). They can also be computationally expensive.

The modeling techniques can be broadly divided into two categories: first-principles modelling and empirical modelling. In case of first-principles modelling, the external driver (i.e., currents in space or their magnetic field on the ground) and ground conductivity are given. The problem is fully controlled and can be applied to simulated external drivers as well as observations, providing a comprehensive way to understand the response of the conducting ground to external driving. In principle, it is possible to solve the electric field, telluric current density, and charge density everywhere, although in practise this is not currently feasible due to the computational expense. In case of empirical modeling, ground-based magnetic field observations are used to estimate the electric field. Information on the external driver and ground conductivity are contained in the observed magnetic field, but further information, such as transfer functions, are needed to estimate the total geoelectric field. The magnetic field can be uniquely separated into an external and internal part, and interpreted in terms of equivalent currents (Juusola et al., 2020). Empirical modelling is often useful in practical GIC applications.

Like any vector field on a sphere, the geoelectric field at the Earth's surface can be separated into divergence-free (DF) and curl-free (CF) parts. The DF electric field is directly associated with temporal variations of the magnetic field via Faraday's law, and can be further separated into an external part due to time-varying ionospheric and magnetospheric currents in space and into an internal part due to telluric currents in the conducting Earth. The CF part is created by electric charges accumulated at conductivity gradients in the ground. There can also be charges in the ionosphere and upper space. However, they are quasi-static and the related electric field is confined between the ionosphere and Earth's surface and is perpendicular to the surface.

Vanhamäki et al. (2013) developed a method for calculating the DF surface electric field using the two-dimensional (2-D) Spherical Elementary Current System (SECS) method (Amm, 1997; Amm and Viljanen, 1999; Pulkkinen et al., 2003a, b; McLay and Beggan, 2010; Weygand et al., 2011; Juusola et al., 2016; Marsal et al., 2017, 2020; Juusola et al., 2020; Laundal

et al., 2021; Vanhamäki and Juusola, 2020; Walker et al., 2023). Their approach was to place a DF 2-D SECS layer in the iono-sphere at 100 km altitude and to determine the SECS amplitudes by fitting the superposed vertical magnetic field components ($B_r$) of the SECSs to the measured vertical magnetic field components on the ground. The horizontal component of the vector potential ($\boldsymbol{A}_h$) was derived by applying the equation for the vector potential of the DF 2-D SECS from Amm and Viljanen (1999). The DF electric field was calculated from the time derivative of the vector potential as

$$\boldsymbol{E}_{DF,h} = -\frac{\partial \boldsymbol{A}_h}{\partial t}. \tag{1}$$

They also discussed the possibility of deriving the CF part of the surface electric field from the known ground surface conduc-tivity and the requirement that the total current is divergence-free.

We will continue the work by Vanhamäki et al. (2013) by deriving a method for calculating the internal and external part of the DF electric field at and above the Earth's surface and the CF electric field at the Earth's surface. Similar to Vanhamäki et al.

(2013), we will utilize the DF 2-D SECSs to calculate the DF electric field, but in addition to the ionospheric DF SECS layer, we will place a second layer just below the Earth's surface, to represent the magnetic field of the telluric currents. A ground conductivity model is not needed for the DF electric field calculation, because the conductivity distribution affects the telluric current distribution, and this is reflected in the magnetic field it produces. Using two layers instead of one allows us to fit all three components of the measured magnetic field when determining the SECS amplitudes, to derive a solution that is valid not

only at the Earth's surface but above it as well, and to separate the internal and external part of the DF electric field.

We will implement and test the calculation for the CF electric field suggested by Vanhamäki et al. (2013). However, we will show that the required assumption of vanishing radial currents is too restrictive and instead derive coefficients based on the first-principles 3-D induction model PGIEM2G (Polynomial Galerkin Integral Equation Modelling in ElectroMagnetic Geophysics) (Kruglyakov and Kuvshinov, 2018) that linearly relate the DF electric field to the horizontal CF electric field at

80 the Earth's surface. PGIEM2G is a scalable 3-D electromagnetic forward modelling code based on a method of volume integral equations with a contracting kernel (Kruglyakov and Kuvshinov, 2018). The resulting geoelectric field will be compared with a) the geoelectric field obtained via fully 3-D simulation using SECS-based ionospheric equivalent currents and PGIEM2G, and b) the geoelectric field modelled via the multi-site transfer function (MSTF) approach (Kruglyakov et al., 2023). The MSTF approach uses transfer functions that relate the horizontal geoelectric field with the horizontal magnetic field in the modelling

region. The MSTFs have been calculated on the basis of electric and magnetic fields produced by PGIEM2G. The MSTF approach partly mitigates inaccuracies in the external source model, improving the accuracy of the resulting geoelectric field. We will also demonstrate how a simple linear relation, based on the same principle as the new method, can be used to relate a local magnetic field time derivative to the local geoelectric field. This application is not as flexible in terms of magnetometer configurations as the SECS method, but it can be used when the magnetometer network is too sparse for the SECS method to

work reliably.

3-D induction modelling codes such as PGIEM2G determine total electric and magnetic fields by solving Maxwell's equa-tions numerically for a given 3-D conductivity model and given inducing source. Such an approach requires relatively long time integration, during which inaccuracies can accumulate, and is sensitive to the available conductivity and source models.

Our approach has the advantage that in addition to the external part of the DF electric field, the internal part of the DF electric field is also derived from measurement. Thus, no conductivity model is needed to determine the DF part of the electric field (information on the conductivity is automatically included in the measured magnetic field) and no time integration is required. However, the sensitivity to the magnetometer coverage remains, as does the dependence of the CF part of the electric field on the conductivity model.

Although first-principles 3-D induction modelling codes are very rigorous and can model ground induction as far as limited magnetic field observations and 3-D ground conductivity models allow, this does not mean that induction is perfectly understood. The geoelectric field is deterministic but highly unpredictable. In other words, theoretically speaking, the electric field can be precisely calculated from a given external source and a known ground conductivity. In practice, however, it is difficult to give any simple rules of how the ground responds to a large variety of different external drivers such as sudden impulses, temporally varying large-scale electrojets, current vortices, etc. As Juusola et al. (2020) show, the internal part of the time derivative of the horizontal ground magnetic field ($d\boldsymbol{H}/dt$) has a much more complex spatial structure than the relatively smooth external $d\boldsymbol{H}/dt$. The contrast becomes even more pronounced when comparing the external driver and the geoelectric field (Marshalko et al., 2023). Separation of the various components of the surface electric field (external and internal parts of the DF electric field and the CF electric field) provides a tool for deepening our understanding, including the combination of external driving and ground conductivity structures that pose the most severe risks for technological conductor networks. Such understanding is particularly useful for space weather services and power grid operators. Furthermore, computationally reasonable methods allow the production of long time series of surface electric field data, which can be used as material for extreme value analysis to better understand the threats of severe space weather events.

The structure of the study is as follows: the new methods are derived and old methods reviewed in Section 2, test results are presented and validated in Section 3, and further applications are discussed in Section 4. The conlusions are summarised in Section 5.

## 2  Methods

In this section, we will first discuss the division of the geoelectric field into DF and CF parts. In Section 2.2, we will derive the method for obtaining the external and internal parts of the DF electric field at and above the Earth's surface from ground magnetic field measurements using DF 2-D SECSs. In Section 2.3.1 we will derive the method for estimating the CF surface electric field from a known DF electric field, surface conductance, and vertical current density using CF 2-D SECS functions. In Section 2.3.1, we will derive the alternative method for estimating the CF surface electric field from a known DF electric field using coefficients derived from first-principles 3-D induction modelling data. In Section 2.4 we will describe how the principles utilized in the DF and CF electric field calculation allow a simple estimation of the total geoelectric field from the time derivative of the magnetic field. Unlike the SECS method, such an approach is rigid in terms of magnetometer station configuration, but it can be convenient when the magnetometer network is too sparse for the SECS method to work reliably.

Finally, in Section 2.5, we will descibe the existing methods, PGIEM2G modelling with the SECS-based source and the MSTF approach, which we use to derive coefficients and to compare with the new models.

## 2.1 Divergence-free and curl-free parts of the geoelectric field

We will approach the modelling problem by separating the different contributions to the geoelectric field. As follows from the Maxwell equations, the electric field $\mathbf{E}$ can be expressed as

$$\mathbf{E} = -\frac{\partial \mathbf{A}}{\partial t} - \nabla \phi, \tag{2}$$

where $\mathbf{A}$ is the vector potential and $\phi$ is the scalar potential. It is possible to require that the vector potential is divergence-free

$$\nabla \cdot \mathbf{A} = 0. \tag{3}$$

This is known as the Coulomb gauge condition (Jackson, 1998). It follows that the scalar potential can be expressed in terms of the charge density $\rho$ as

$$\phi(\mathbf{r}, t) = \frac{1}{4\pi\epsilon_0} \int_V \frac{\rho(\mathbf{r}', t)}{|\mathbf{r} - \mathbf{r}'|} dV', \tag{4}$$

where $\epsilon_0$ is the vacuum permittivity. Contrary to the often applied Lorenz gauge, the time $t$ is not retarded, but the scalar potential is formally identical to the static case.

In the Coulomb gauge, the vector potential is given by the divergence-free (DF) part of the current density $\mathbf{J}$

$$\mathbf{A}(\mathbf{r}, t) = \frac{\mu_0}{4\pi} \int_V \frac{\mathbf{J}_{DF}(\mathbf{r}', t - |\mathbf{r} - \mathbf{r}'|/c)}{|\mathbf{r} - \mathbf{r}'|} dV', \tag{5}$$

where $\mu_0$ is the vacuum permeability and $c$ is the speed of light. Note that here the time is retarded.

The power of the Coulomb gauge is that it automatically separates the electric field into a curl-free (CF) static part and a divergence-free inductive part

$$\mathbf{E} = \mathbf{E}_{CF} + \mathbf{E}_{DF}, \tag{6}$$

where

$$\mathbf{E}_{CF} = -\nabla \phi, \ \mathbf{E}_{DF} = -\frac{\partial \mathbf{A}}{\partial t}, \tag{7}$$

and $\mathbf{E}_{CF}$ is produced only by charges and $\mathbf{E}_{DF}$ only by divergence-free currents as explicitly shown above.

## 2.2 DF electric field of a DF 2-D SECS with a time-varying amplitude

The DF part of the electric field ($\boldsymbol{E}$) due to a time-varying magnetic field ($\boldsymbol{B}$) can be solved from Faraday's law

$$\nabla \times \boldsymbol{E}_{DF} = -\frac{\partial \boldsymbol{B}}{\partial t}. \tag{8}$$

DF 2-D SECSs form a complete set of basis functions for representing ground magnetic field variations in terms of external and internal equivalent currents (Vanhamäki and Juusola, 2020). The time-varying sheet current density of the DF 2-D SECS (Amm, 1997) is

$$\boldsymbol{J}_{DF}(t) = \frac{I_{DF}(t)}{4\pi R} \cot\left(\frac{\theta'}{2}\right) \hat{\boldsymbol{e}}_{\phi'}, \tag{9}$$

where $I_{DF}(t)$ is the time-varying amplitude in A, $R$ is the radius of the current sheet, and $\theta'$ is the colatitude in the SECS coordinates, where $\theta' = 0$ at the SECS pole. Conversion to general coordinates, e.g., geographic, follows the regular practise of the 2-D SECS method (Vanhamäki and Juusola, 2020). It should be noted that although the resulting external DF current density can only be interpreted as the DF part of the ionospheric horizontal current density at high latitudes, where the ambient magnetic field lines can be approximated as radial, it can still describe the ground magnetic field everywhere below the ionosphere (e.g., Juusola et al., 2023a, b). Thus, the applicability of the DF electric field calculation method is not limited to high latitudes.

In the SECS coordinates, the time derivative of the DF 2-D SECS magnetic field only has $r$ and $\theta'$ components (Amm and Viljanen, 1999)

$$\frac{\partial \boldsymbol{B}}{\partial t} = \frac{\partial B_r}{\partial t} \hat{\boldsymbol{e}}_r + \frac{\partial B_{\theta'}}{\partial t} \hat{\boldsymbol{e}}_{\theta'}, \tag{10}$$

where

$$\frac{\partial B_r}{\partial t} = \frac{\partial I_{DF}}{\partial t} \frac{\mu_0}{4\pi} \frac{1}{r} \left( \frac{R}{\sqrt{r^2 - 2rR\cos\theta' + R^2}} + \left\{ \begin{array}{c} -1 \\ -R/r \end{array} \right\} \right) \begin{array}{l} \text{when } r < R \\ \text{when } r \geq R \end{array} \tag{11}$$

and

$$\frac{\partial B_{\theta'}}{\partial t} = -\frac{\partial I_{DF}}{\partial t} \frac{\mu_0}{4\pi} \frac{1}{r\sin\theta'} \left( \frac{r - R\cos\theta'}{\sqrt{r^2 - 2rR\cos\theta' + R^2}} + \left\{ \begin{array}{c} \cos\theta' \\ -1 \end{array} \right\} \right) \begin{array}{l} \text{when } r < R \\ \text{when } r > R \end{array}. \tag{12}$$

This follows from neglecting the displacement current, as is usual in geoelectromagnetism. However, it should be noted that the displacement current may play a role in producing ground-based magnetic field signatures that do not cause a geoelectric field on the ground (Brändlein et al., 2012). Due to the geometry of the DF SECS magnetic field, the corresponding induced electric field only has a $\phi'$ component (see also the vector potential derivation in Amm and Viljanen (1999))

$$\boldsymbol{E}_{DF} = E_{DF,\phi'} \hat{\boldsymbol{e}}_{\phi'}, \tag{13}$$

which can be solved from the integral form of Equation 8

$$\oint_{\partial A} \boldsymbol{E}_{DF} \cdot d\boldsymbol{l} = -\frac{\partial}{\partial t} \int_A \boldsymbol{B} \cdot d\boldsymbol{S} \tag{14}$$

by calculating the change in the magnetic flux through the spherical cap surface defined by a field line of $E_{DF,\phi'}$ (Figure 1)

$$2\pi r \sin\theta' E_{DF,\phi'} = -r^2 \int_0^{\theta'} d\theta \sin\theta \int_0^{2\pi} d\phi \frac{\partial B_r}{\partial t}. \tag{15}$$

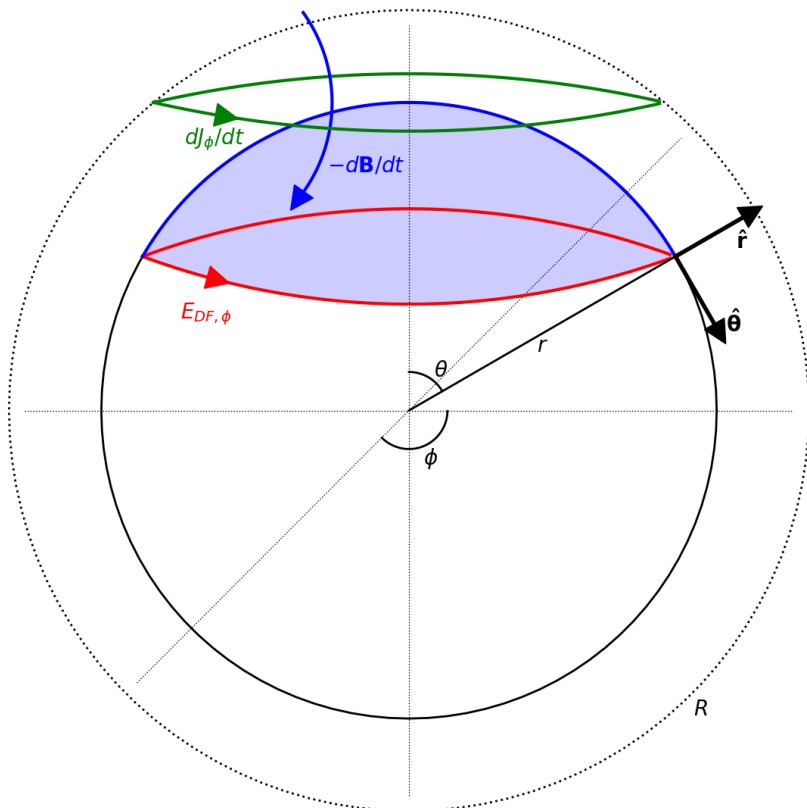

**Figure 1.** Geometry of the derivation of the divergence-free (DF) electric field ($E_{DF,\phi}$) of a DF 2-D Spherical Elemenatry Current System (SECS) with a time-varying amplitude. $J_\phi$ indicates the SECS current density, $\boldsymbol{B}$ its magnetic field, and $d/dt$ the time derivative. $r$, $\theta$, and $\phi$ are spherical coordinates (radius, co-latitude, and longitude) and $\hat{\boldsymbol{r}}$, $\hat{\boldsymbol{\theta}}$, $\hat{\boldsymbol{\phi}}$ the corresponding unit vectors.

Carrying out the calculation yields

$$E_{DF,\phi'} = -\frac{\partial I_{DF}}{\partial t}\frac{\mu_0}{4\pi}\frac{1}{r\sin\theta'}\left(\sqrt{r^2 - 2rR\cos\theta' + R^2} + \begin{Bmatrix} r\cos\theta' - R \\ R\cos\theta' - r \end{Bmatrix}\right) \begin{array}{l} \text{when } r < R \\ \text{when } r \geq R \end{array} \tag{16}$$

which is equal to the negative time derivative of the vector potential (Eq. 1) given by Amm and Viljanen (1999) for the DF

SECS when $r < R$. Comparison of Eq. 16 with Eq. 9 confirms that the induced electric field opposes the temporal change of the DF current, as expected from Lenz's law.

Figure 2 shows co-latitude profiles of $\partial J_\phi/\partial t$, ground $\partial B_r/\partial t$, ground $\partial B_\theta/\partial t$, and ground $E_{DF,\phi}$ for two different cases: $J_\phi$ at 90 km altitude (blue curves, external contribution) and oppositely directed $J_\phi$ at $2\times90$ km depth (red curves, internal contribution), to mimic a simple case of ionospheric and telluric currents. Both have the same amplitude $|\partial I_{DF}/\partial t| = 10\,\text{kA/s}$.

The sum of the external and internal $\partial B_r/\partial t$, $\partial B_\theta/\partial t$, and $E_\phi$ on the ground are shown by the black curves. While the current density is concentrated near the SECS pole at $\theta = 0$, the magnetic field is more spread out to lower latitudes, and the DF electric

field even more so. On the ground, the horizontal magnetic field components $B_\theta$ from the oppositely directed ionospheric and telluric currents strengthen each other, while the vertical components $B_r$ tend to cancel each other out. The DF electric fields also tend to cancel each other out, resulting in a much weaker total ground DF electric field than either the original ionospheric or telluric contribution.

Brändlein et al. (2012) discussed a waveguide transmission, where the wave mode on the ground has a non-zero horizontal magnetic field component but a zero horizontal electric field. In the vertical direction such a wave mode is expected to have a zero magnetic and non-zero electric field components. The SECS reconstruction is able to reproduce such a magnetic field as a superposition of the magnetic fields of ionospheric and telluric equivalent currents. Because the vertical magnetic field is zero, $\boldsymbol{E}_{DF}$ would also be zero.

Although the combination of ionospheric and telluric DF current densities always produces a DF electric field that has a zero vertical component between the ionspheric and telluric equivalent current sheets, this does not necessarily mean that $\boldsymbol{E}_{DF}$ cannot have a vertical component in this region. This issue was investigated in detail by Pirjola and Viljanen (1998). In addition to the parts described by the DF current densities, the 3D current distributions in the ionosphere and in the ground include a part that has a zero magnetic field between the ionosphere and ground surface. However, the corresponding vector potential $\boldsymbol{A}$ may not be zero, although $\nabla \times \boldsymbol{A}$ must be zero. The corresponding DF electric field is a Laplace field that has its sources above the ionosphere and inside the ground. A similar Laplace electric field could also be produced with electric charges in these regions. According to the results by Pirjola and Viljanen (1998), valid up to neglecting the displacement current, any horizontal part of this DF field is cancelled by charges accumulated on the ground surface, leaving only an insignificant vertical component. Thus this part of the induction process does not drive any GIC.

The time derivative of the SECS amplitude can be estimated from the known amplitudes $I_{DF}$ as

$$\frac{\partial I_{DF}}{\partial t} = \frac{I_{DF}(t) - I_{DF}(t - \Delta t)}{\Delta t}, \tag{17}$$

where $\Delta t$ is the time step of the data. When both the ionospheric equivalent current amplitudes at 90 km altitude and telluric equivalent current amplitudes at 1 m depth have been determined from ground magnetic field observations (Juusola et al., 2016, 2020; Vanhamäki and Juusola, 2020), Eq. 16 can be used to estimate the ionospheric DF current contribution to $\boldsymbol{E}_{DF}$ everywhere and telluric current contribution to $\boldsymbol{E}_{DF}$ at and above the Earth's surface.

## 2.3 CF electric field at the Earth's surface driven by the DF electric field

### 2.3.1 Approach A: CF SECS

The DF electric field in the conducting Earth has an external and an internal component. The external component is produced by time-varying ionospheric and magnetospheric current systems, and the internal component by time-varying 3-D telluric currents. This electric field drives the electric currents in the Earth. Whenever the current driven by the DF electric field crosses a conductivity gradient, there is divergence of the current. Because the total current must be divergence-free (follows from neglecting the displacement current, as usual in geoelectromagnetism), a curl-free (CF) electric field is set up by redistribution

of electric charges in the Earth to satisfy this condition. The divergence of the total current can be written as

$$\nabla \cdot \boldsymbol{j} = \nabla \cdot (\sigma \boldsymbol{E}_{DF} + \sigma \boldsymbol{E}_{CF}) = \nabla \sigma \cdot \boldsymbol{E}_{DF} + \sigma \nabla \cdot \boldsymbol{E}_{DF} + \nabla \sigma \cdot \boldsymbol{E}_{CF} + \sigma \nabla \cdot \boldsymbol{E}_{CF},\tag{18}$$

where $\sigma(r,\theta,\phi)$ is the known, location-dependent, electrical conductivity. The DF electric field is a superposition of the external and internal contribution and satisfies

$$\nabla \cdot \boldsymbol{E}_{DF} = 0.\tag{19}$$

We will only consider the surface layer of the conducting Earth and model it as an infinitely thin conducting sheet located at the Earth's surface. Equation 18 then becomes

$$\nabla \Sigma \cdot \boldsymbol{E}_{CF} + \Sigma \nabla \cdot \boldsymbol{E}_{CF} = -\nabla \Sigma \cdot \boldsymbol{E}_{DF} + j_r,\tag{20}$$

where $\Sigma$ is the conductance of the thin sheet, obtained by vertical integration of the conductivity in the surface layer, and $j_r$ is the vertical current density.

We use CF SECS functions to obtain $\boldsymbol{E}_{CF}$ when $\Sigma$ and $\boldsymbol{E}_{DF}$ are known. The electric field of a CF SECS (Vanhamäki and Juusola, 2020) is

$$\boldsymbol{E}_{CF} = \frac{Q_{CF}}{4\pi R} \hat{e}_{\theta'} \left\{ \begin{array}{l} \cot\left(\frac{\theta'}{2}\right) \\ \tan\left(\frac{\theta'}{2}\right)\cot^2\left(\frac{\theta_0}{2}\right) \end{array} \right. \begin{array}{l} \text{when } \theta' \geq \theta_0 \\ \text{when } \theta' < \theta_0 \end{array},\tag{21}$$

where $Q_{CF}$ is the amplitude of the CF SECS in V (i.e., $Q_{CF}$ is charge divided by $\epsilon_0$), $R$ is the radius of the surface, and $\theta'$ is the colatitude in the SECS coordinates. The source at the elementary current system's pole is spread uniformly inside a spherical cap of width $\theta_0$. The divergence of $\boldsymbol{E}_{CF}$ is given by

$$\nabla \cdot \boldsymbol{E}_{CF} = \frac{\rho}{\epsilon_0} = -\frac{Q_{CF}}{4\pi R^2} + \left\{ \begin{array}{l} 0 \\ \frac{Q_{CF}}{A} \end{array} \right. \begin{array}{l} \text{when } \theta' \geq \theta_0 \\ \text{when } \theta' < \theta_0 \end{array},\tag{22}$$

where $\rho$ is the charge density and $A$ is the area of the spherical cap defined by $\theta_0$

$$A = 2\pi R^2 (1 - \cos\theta_0).\tag{23}$$

The equations above apply to a single SECS pole. The total CF electric field in the grid cell $i$ is a superposition of the electric fields of all the CF SECSs. Eq. 20 can now be written as

$$\sum_j \left[ \nabla \Sigma_{\theta,i} g_{\theta,i,j} + \nabla \Sigma_{\phi,i} g_{\phi,i,j} - \Sigma_i \frac{1}{4\pi R^2} \right] Q_{CF,j} + \Sigma_i \frac{Q_{CF,i}}{A_i} = -\nabla \Sigma_{\theta,i} E_{DF\theta,i} - \nabla \Sigma_{\phi,i} E_{DF\phi,i} + j_{r,i},\tag{24}$$

where we have denoted the geometry-dependent components of $\boldsymbol{E}_{CF}$ by $g$. By assuming that the radial current either vanishes ($j_{r,i} = 0$) or is otherwise known, the amplitudes $Q_{CF,i}$ can be determined from the matrix equation corresponding to Eq. 24. No time-integration of currents is needed, because the internal contribution to $\boldsymbol{E}_{DF}$ is obtained from measurements. Thus, only two time steps of data are needed to determine the geoelectric field at the Earth's surface for a given epoch.

### 2.3.2 Approach B: Linear coefficients relating $E_{CF}$ to $E_{DF}$

The CF 2-D SECS-based method described above assumes that the vertical current density in the surface layer is known. We will demostrate below that this assumption is too restrictive. As an alternative approach, we assume that the CF electric field is approximately proportional to the driving DF electric field

$$E_{CF,x} = k_{xx}E_{DF,x} + k_{xy}E_{DF,y} \tag{25}$$

$$E_{CF,y} = k_{yy}E_{DF,y} + k_{yx}E_{DF,x}. \tag{26}$$

This formulation resembles that of magnetotelluric transfer functions, which define the frequency dependent linear relationship (impedance tensors) between components of the electric field and horizontal components of the magnetic field variations measured at a given location. Our formulation does not depend on frequency but is given in the time domain. This is possible, because the internal part of the DF electric field is obtained from measurements and we only need a linear relationship for describing the CF part of the electric field from the known DF part. The assumed linear relation contains a local plane wave assumption, i.e., we assume that the spatial structures of the DF electric field are larger than the range of the relevant charge accumulations that contribute to the local CF field. A more general relationship would be of the form ($\boldsymbol{E}_{CF}$ values) = matrix · ($\boldsymbol{E}_{DF}$ values), with all grid points collected in the matrices. However, the simple the simple assumptions of Eq. 25–26 appear to be sufficient, as we will demonstrate below. In principle, it should be enough to to determine the time-independent coefficients from a single active interval of modelled geoelectric field data using. The most obvious problem would arise from a case where a persistent small-scale DF electric field structure would bias the coeffcients at some locations. However, the chaotic nature of the time derivative of the magnetic field (Kellinsalmi et al., 2022) makes such a case unlikely, as long as the event chosen is sufficiently long and active.

### 2.4 Relating $E$ to $\frac{\partial B}{\partial t}$

The linear relation between the DF electric field and $\frac{\partial \boldsymbol{B}}{\partial t}$ on the one hand, and the linear relation between the CF and DF electric field on the other hand, implies that the geoelectric field at location $\boldsymbol{r}$ can be estimated from the magnetic field observations at locations $\boldsymbol{r}_k^{obs}$ as

$$\boldsymbol{E}(\boldsymbol{r},t) = \sum_{k=1}^{N_{obs}} Q(\boldsymbol{r},\boldsymbol{r}_k^{obs})\frac{\partial}{\partial t}\boldsymbol{B}(\boldsymbol{r}_k^{obs},t), \tag{27}$$

where $Q(\boldsymbol{r},\boldsymbol{r}_k^{obs})$ are $2 \times 3$ time-independent matrices. The $Q$ matrices can be determined using, e.g., PGIEM2G simulations, and bypass the need for the SECS expansion. However, unlike the SECS expansion, the $Q$ matrices are rigid in terms of station configuration and introduce dependence on the conductivity model to the DF electric field. Determining the matrices will be a topic for a separate study.

The SECS method requires a relatively dense regional magnetometer network. For a single location with a limited time series of electric field observations and a more extended time series of magnetic field observations, the electric field time series

 can be approximated by defining the coefficients $q$

$$E_x = q_{xx}\frac{\partial B_x}{\partial t} + q_{xy}\frac{\partial B_y}{\partial t} + q_{xz}\frac{\partial B_z}{\partial t} \tag{28}$$

$$E_y = q_{yx}\frac{\partial B_x}{\partial t} + q_{yy}\frac{\partial B_y}{\partial t} + q_{yz}\frac{\partial B_z}{\partial t}. \tag{29}$$

An applicable case might occur when a magnetotelluric measurement is carried out close to the location of a semi-permanent magnetometer station.

## 2.5 Existing methods

### 2.5.1 PGIEM2G

PGIEM2G (Polynomial Galerkin Integral Equation Modelling in ElectroMagnetic Geophysics) (Kruglyakov and Kuvshinov, 2018) is a scalable 3-D electromagnetic forward modelling code based on a method of volume integral equations with a contracting kernel (Pankratov and Kuvshinov, 2016; Kruglyakov and Kuvshinov, 2018). The conductivity model used by PGIEM2G comprises a 3-D part (upper 60 km) and a 1-D conductivity profile (below 60 km) from Kuvshinov et al. (2021). The 3-D part is based on the SMAP model (Korja et al., 2002), covers an area of $2,550 \times 2,550$ km$^2$, and consists of three layers of laterally variable conductivity of 10, 20, and 30 km thicknesses. The lateral discretization of the model is $512 \times 512$ cells. This model was also exploited in Marshalko et al. (2021), Kruglyakov et al. (2022), Marshalko et al. (2023), and Kruglyakov et al. (2023). The external source used in the PGIEM2G simulation is obtained using the SECS method (cf. Marshalko et al., 2021, 2023). As the integral equation approach is invoked, the model space is infinite.

Calculations in PGIEM2G code are carried out in Cartesian geometry for planar Earth. To make the transition from spherical to Cartesian geometry, a map projection (Transverse Mercator with the latitude and longitude of the true origin at 50$^o$N and 25$^o$E, respectively) and interpolation onto a regular grid (about $5 \times 5$ km) is first performed for the SMAP and inducing source data. Then an inverse transformation and interpolation onto a regular grid is carried out for the resulting electric and magnetic fields to obtain the data in spherical geometry. The final spatial resolution of electric and magnetic fields is $0.03° \times 0.07°$ in latitudinal and longitudinal directions, correspondingly. The discrepancy between the SMAP grid used in some of our calculations and the denser grid used in PGIEM2G is not considered an issue in our qualitative comparison. In the one case when we wish to insert part of the PGIEM2G data directly into our model (section 3.2), nearest-neighbour interpolation is performed.

### 2.5.2 MSTF

The multi-site transfer function (MSTF) approach proposed by Kruglyakov et al. (2023) is based on the use of transfer functions that linearly relate the horizontal geoelectric field $\boldsymbol{E}_h = (E_x, E_y)$ at any location in the modelling region with the horizontal magnetic field $\boldsymbol{B}_h = (B_x, B_y)$ at (fixed) $N$ locations in frequency domain

$$E_h(r,\omega;\sigma) = \sum_{k=1}^{N} \Lambda_k(r,r_k,\omega;\sigma)B_h(r_k,\omega;\sigma). \tag{30}$$

Here $r$ is a position vector, $\omega$ is angular frequency, $\sigma(r)$ is the conductivity distribution in the modelling region, and $\Lambda_k$ are the following $2 \times 2$ matrices

$$\Lambda_k = \begin{pmatrix} \Lambda_{xx,k} & \Lambda_{xy,k} \\ \Lambda_{yx,k} & \Lambda_{yy,k} \end{pmatrix}. \tag{31}$$

Estimation of the elements of matrices $\Lambda_k$, $k = 1,2,\ldots,N$ at a given frequency $\omega$ and conductivity model $\sigma$ is performed by use of the modelled fields $B^{(l)}(r_k,\omega;\sigma)$ and $E^{(l)}(r,\omega;\sigma)$ corresponding to $l$-th spatial modes $j_l(r)$ describing the external source

$$j^{\text{ext}}(r,t) = \sum_{l=1}^{L} c_l(t)j_l(r), \tag{32}$$

where $L$ stands for a number of spatial modes (see, Kruglyakov et al., 2022). After these fields are calculated for a given computational model $\sigma(r)$ they are substituted into Eq. (30) to form the linear system for elements of $\Lambda_k$ that, in turn, is solved by means of regularized ordinary least squares approach.

Once elements of $\Lambda_k$ are estimated at a predefined number of frequencies, and, assuming that time-series of the observed magnetic field are provided at uniform time steps $\delta_t$, $E_h(r,t_i)$ for any given time instant $t_i = i\delta_t$ and a given location $r$ is calculated using the numerical scheme described in Kruglyakov et al. (2022) and Marshalko et al. (2023)

$$E_h(r,t_i;\sigma) = \sum_{n=0}^{N_t}\sum_{k=1}^{N} \Lambda^{(n)}(r,r_k,T_\Lambda;\sigma)B_h^{\text{obs}}(r_k,t_i - n\delta_t), \tag{33}$$

where $B_h^{\text{obs}}$ stands for horizontal components of the observed field in the corresponding locations, $T_\Lambda$ is the so-called "memory", and $N_t = T_\Lambda/\delta_t$. The detailed methodology of computing $\Lambda^{(n)}(r,r_k,T_\Lambda;\sigma)$ from $\Lambda(r,r_k,\omega;\sigma)$ is provided in Kruglyakov et al. (2022, 2023).

The MSTF used in this manuscript were calculated using the same conductivity model as in previous section and the 34 spatial modes identified previously by Marshalko et al. (2023) via the time-domain Principal Component Analysis of the SECS ionospheric equivalent current in Fennoscandia during three days of the Halloween geomagnetic storm (29–31 October 2003). The corresponding forward simulations were carried out via PGIEM2G.

The calculations of geoelectric field during the 7–8 September geomagnetic storm via MSTF approach were performed using $T_\Lambda = 15$ min (cf., Kruglyakov et al., 2022, 2023) and the data from $N = 15$ IMAGE magnetometer network stations (ABK, AND, DOB, HAN, IVA, KEV, KIR, NUR, OUJ, RVK, SOD, SOR, TAR, TRO, and UPS).

## 3 Results and validation

Marshalko et al. (2021) have used PGIEM2G to model the 3-D geoelectric field during the September 7–8 2017 storm. They show the resulting horizontal geoelectric field for two epochs, 23:16 UT and 23:52 UT. We use the epoch 23:16 UT to test our method. Corresponding illustrations for the epoch 23:52 UT can be found in the Appendix. Marshalko et al. (2021) ran their simulation using inducing source data with $1 \, \mathrm{min}$ temporal resolution as an input, but we compare our results with a rerun that used $10 \, \mathrm{s}$ data.

### 3.1 DF electric field

We have used $10 \, \mathrm{s}$ ground magnetic field measurements from the International Monitor for Auroral Geomagnetic Effects (IMAGE) magnetometers (Juusola et al., 2024). After correcting the data for any erroneous spikes and jumps, a 10-day sliding median baseline was subtracted from the data. DF 2-D SECS poles were placed in the IMAGE region on uniform grids with $0.5^o$ latitude and $1^o$ longitude resolution at $1 \, \mathrm{m}$ depth and at $90 \, \mathrm{km}$ altitude, and their amplitudes were determined by fitting

the superposed magnetic field of the SECSs to the three components of the measured magnetic field (Vanhamäki and Juusola, 2020). The time derivative of the SECS amplitudes was derived using Eq. 17 and the DF electric field using Eq. 16 combined with the regular coordinate manipulations of the SECS method (Vanhamäki and Juusola, 2020). Because in this application of the 2-D SECS method we are only interested in $\partial \boldsymbol{B}/\partial t$ and not $\boldsymbol{B}$, it would also be possible to skip the baseline subtraction and directly fit $\partial \boldsymbol{B}/\partial t$ with the DF 2-D SECS functions.

Figure 3 shows the resulting external, internal, and total time derivative of the ground magnetic field ($\partial \boldsymbol{B}/\partial t$) and the DF electric field ($\boldsymbol{E}_{DF}$) and its curl (Eq. 8) on 7 Sep 2017 at 23:16 UT. $\boldsymbol{E}_{DF}$ is smoother than $\partial \boldsymbol{B}/\partial t$ but peaks in the same general area as the horizontal part of $\partial \boldsymbol{B}/\partial t$, as expected. Fig. 3 also demonstrates that the internal parts have much more pronounced spatial variability than the relatively smoothly varying external parts. External sources are located more than $\sim 100 \, \mathrm{km}$ from the ground points, whereas internal sources (telluric currents) are concentrated close to the surface and strongly modulated

by conductivity gradients. The direction of the internal part of $\boldsymbol{E}_{DF}$ is opposite to that of the external part of $\boldsymbol{E}_{DF}$, and has a slightly weaker amplitude. As a result, the total $\boldsymbol{E}_{DF}$ has a clearly weaker amplitude and different distribution than the external and internal parts. The overall amplitude level of the DF electric field of hundreds of $\mathrm{mV/km}$ is in agreement with the theoretical examples of Vanhamäki et al. (2013). The curl $(\nabla \times \mathbf{E})_r$ equals $-\partial B_r/\partial t$ and $\partial \boldsymbol{B}/\partial t$ is conserved. It should be noted that this does not guarantee that the separation into internal and external parts is entirely correct. The two sources may

get mixed to some degree, especially in areas where the magnetometer coverage is sparse (Juusola et al., 2020).

### 3.2 CF electric field

Similar to Marshalko et al. (2021), we have used the SMAP model (Korja et al., 2002) to estimate the ground conductivity $\sigma$. SMAP provides the conductivity with a $5'$ latitude and $5'$ longitude resolution. The conductivity distribution in the top 0–10 km layer is displayed in Figure 4.

We have determined the CF SECS amplitudes on a grid where the conductance, its gradient, and grid cell area are obtained

with the finite difference approach

$$\Sigma_{i,j} = \sigma_{i,j} \cdot 10 \text{ km} \tag{34}$$

$$\Sigma_{i+\frac{1}{2},j+\frac{1}{2}} = \frac{\Sigma_{i,j}+\Sigma_{i+1,j}+\Sigma_{i,j+1}+\Sigma_{i+1,j+1}}{4} \tag{35}$$

$$\nabla\Sigma_{\theta,i+\frac{1}{2},j+\frac{1}{2}} = \frac{1}{2} \cdot \left( \frac{\Sigma_{i+1,j}-\Sigma_{i,j}}{R_E\Delta\theta} + \frac{\Sigma_{i+1,j+1}-\Sigma_{i,j+1}}{R_E\Delta\theta} \right) \tag{36}$$

$$\nabla\Sigma_{\phi,i+\frac{1}{2},j+\frac{1}{2}} = \frac{1}{2} \cdot \left( \frac{\Sigma_{i,j+1}-\Sigma_{i,j}}{R_E\sin\theta_{\phi,i+\frac{1}{2},j+\frac{1}{2}}\Delta\phi} + \frac{\Sigma_{i+1,j+1}-\Sigma_{i+1,j}}{R_E\sin\theta_{\phi,i+\frac{1}{2},j+\frac{1}{2}}\Delta\phi} \right) \tag{37}$$

$$A_{i+\frac{1}{2},j+\frac{1}{2}} = 2R_E^2\Delta\phi\sin\theta_{i+\frac{1}{2},j+\frac{1}{2}} \sin\frac{\Delta\theta}{2} \tag{38}$$

$$j_{r,i+\frac{1}{2},j+\frac{1}{2}} = 0. \tag{39}$$

Here, $R_E$ is the Earth radius, $\Delta\theta$ and $\Delta\phi$ are the colatitude and longitude resolution of the original conductivity grid, $i$ is an

index of the colatitude grid and $j$ an index of the longitude grid. As we do not have any information of the vertical current

density, it is assumed to vanish. The SECS polar cap angle $\theta_0$ was calculated from the grid cell area using Eq. 23. The final CF

electric field was evaluated at the original conductance grid, i.e., at grid points $(i, j)$.

Because of the computational limitations of a regular laptop, some optimization of the CF electric field calculation was

needed to process the full IMAGE area: We split the conductivity model area into 240 pieces with an extent of $2^o$ in latitude

with $\pm 2^o$ padding and $1.9^o$ in longitude with $\pm 4^o$ padding. The full area was assembled by fitting these 240 subareas together.

This approximation did not extend to the DF electric field calculation, for which the full area was utilized. Naturally, the best

approach would be to utilize more powerful computing resources, so that no optimization would be needed at all. For our

simple test the appproximation sufficies.

Figure 5 displays the resulting charge density (5a), amplitude of the horizontal part of the CF electic field $\boldsymbol{E}_{CF}$ (5b), and total

electric field $\boldsymbol{E}_{DF} + \boldsymbol{E}_{CF}$ (5c) for the same epoch as Fig. 3. The edges of the computational subareas can be detected in some

regions, indicating that larger areas would be better. As expected, charges accumulate at conductivity gradients, producing a

total electric field that is significantly modulated compared to the DF electric field of Fig. 3f. Especially, the tendency for higher

electric field in areas of lower conductivity and lower electric field in areas of higher conductivity is more pronounced.

### 3.3 Comparison with PGIEM2G

In this section, we compare the electric field modelled using the DF and CF SECS methods with the electric field modelled

using PGIEM2G (Kruglyakov and Kuvshinov, 2018). Figure 6a shows the amplitude of the horizontal surface electric field

modelled using PGIEM2G for the same epoch as Fig. 5c. The rest of the panels show the DF part of the electric field and its

curl (6b), CF part of the electric field (6c), divergence of the electric field (6d), and divergence of the horizontal current density

divided by conductance (6e). We will come back to Fig. 6f later.

The 2-D SECS method can also be used to separate any vector field on a sphere into its DF and CF parts (Amm, 1997). We

have utilized it to separate the total electric field produced by PGIEM2G. The DF part of the electric field was obtained using

the modelled curl $(\nabla \times \boldsymbol{E})_r = -\frac{\partial B_r}{\partial t}$ to estimate DF 2-D SECS amplitudes $I_{DF}$ on the PGIEM2G grid

$$(\nabla \times \boldsymbol{E})_{r,i} = \frac{I_{DF,i}}{A_i} - \sum_j \frac{I_{DF,j}}{4\pi R^2} \tag{40}$$

$$I_{DF,i+\frac{1}{2},j+\frac{1}{2}} \approx A_{i+\frac{1}{2},j+\frac{1}{2}} (\nabla \times \boldsymbol{E})_{r,i+\frac{1}{2},j+\frac{1}{2}}. \tag{41}$$

The CF part of the electric field was obtained by subtracting the DF part of the electric field from the total electric field. The
divergence of the electric field $(\nabla \cdot \boldsymbol{E})$ was estimated using finite difference calculation.

Comparison of the PGIEM2G (Fig. 6a) and SECS (Fig. 5c) results reveals that, unlike PGIEM2G, the SECS method does not produce a narrow band of intense CF electric field on the side of the low conductance at conductance gradients but enhances the electric field across the entire area of lower conductance. The SECS method assumes that vertical current density is zero, but clearly this is not a valid assumption, as can be seen by examining the divergence of the horizontal current density in
Fig. 6e. Comparison of Fig. 6e with the divergence of the electric field in Fig. 6d confirms that vertical currents are clearly responsible for producing the second layer of charges near conductivity gradients that largely confine the CF electric field near the gradients.

Another possible source of discrepancy between PGIEM2G and the SECS method is the internal part of the DF electric field. Whereas the SECS method reconstructs it from ground-based geomagnetic measurements, PGIEM2G models it based on the
3-D conductivity of SMAP. While sparse magnetometer coverage affects the DF electric field of the SECS method, inaccuracies in the conductivity model affect PGIEM2G. The difference can be seen by comparing the respective DF electric fields in Fig. 3f and Fig. 6b. PGIEM2G produces small-scale structures that cannot be resolved with the operational magnetometers. On the other hand, the curl of the electric field from the SECS method corresponds to the measured $-\partial B_r/\partial t$ at all IMAGE stations whereas for PGIEM2G this is not true, as can be seen by comparing with $\partial B_r/\partial t$ in Fig. 3c.
We test the above conclusions by running the SECS CF electric field calculation with PGIEM2G's DF electric field and vertical current density. The resulting divergence of the electric field is given in Fig. 5d for comparison with the CF SECS method and repeated in Fig. 6f for comparison with PGIEM2G. Now there is a much better correspondence between the CF SECS result in Fig. 6f and PGIEM2G result in Fig. 6d. Thus, we conclude that without knowledge of the vertical current density, the SECS method cannot reliably estimate the CF part of the electric field. The DF part of the electric field, however,
can be considered reliable, at least in areas where the magnetometer coverage is good.

### 3.4   Coefficients relating $\mathbf{E}_{CF}$ to $\mathbf{E}_{DF}$

As we cannot use the CF SECS method to estimate the CF part of the electric field, we try an alternative approach: The method described in Section 2.4, which assumes that the CF electric field is linearly proportional to the driving DF electric field. Figure 7 shows the coefficients derived using the CF and DF part of PGIEM2G's horizontal surface electric field between
7 September 2017 23:00 UT and 8 September 2017 01:00 UT. The standard deviation (STD) errors of the coefficients are displayed in Figure 8. These are small compared to the values of the coefficients.

In order to examine the validity of Eq. 25–26, Figure 9 shows a time series of the CF electric field at SOD between 7 September 2017 23:00 UT and 8 September 2017 01:00 UT. The original PGIEM2G-modelled value is shown in black and the

approximation based on Eq. 25–26 is shown in red. There is good correspondence between the original curve and approxima-
tion. Furthermore, Figure 10 shows maps of the CF part of the horizontal electric field modelled by PGIEM2G on 7 Sep 2017
at 23:16:00 UT (10a), the CF electric field approximated using Eq. 25–26 with the coefficients from Fig. 7 and DF electric
field from PGIEM2G as shown in Fig. 6b (10b), and the difference between the original and approximated CF field (10c). The
differences are generally not very large, indicating that this approximation could be used to estimate the CF electric field from
a known DF electric field. However, it is worth noting that the results are obtained for Fennoscandia and may be not valid in
case of different conductivity distributions.

Fig. 5e shows a map of the CF electric field approximated using Eq. 25–26 with the coefficients from Fig. 7 and the IMAGE-
based DF electric field from Fig. 3f. Fig. 5f shows the corresponding total electric field, i.e., the sum of the DF electric field
in Fig. 3f and the CF electric field in Fig. 5e. Now the CF and total electric field structures are similar to those produced
by PGIEM2G (Fig. 6c) although there are some differences due to the DF electric field (Fig. 3f and Fig. 6b). Furthermore,
Figure 11 shows $E_x$ and $E_y$ at SOD between 7 September 2017 23:00 UT and 8 September 2017 01:00 UT. PGIEM2G
modelling is shown in black, DF SECS modelling based on IMAGE magnetic field with the CF part obtained using the
coefficients from Fig. 7 in red, and DF SECS modelling based on PGIEM2G magnetic field with the CF part obtained using
the coefficients from Fig. 7 in green. We have also added estimates obtained via the MSTF (Kruglyakov et al., 2023) approach
(blue curves). In Fig. 11, the SECS modelling based on IMAGE data shows peaks of higher amplitude than either PGIEM2G
or the MSTF approach. Because the SECS modelling based on PGIEM2G magnetic field data has a much better agreement
with PGIEM2G, the peaks in the IMAGE-based SECS are clearly due to the difference in the internal DF electric field.

Finally, Figure 12 shows a time series of the electric field as a sum of the external DF, internal DF, and CF parts, at the
location of the magnetometer station SOD between 7 September 2017 23:00 UT and 8 September 2017 01:00 UT. The DF
electric field has been modelled using the DF SECS method and IMAGE data, and the CF electric field has been obtained using
the coefficients from Fig. 7. As can be seen from Fig. 4, SOD is located close to a conductivity gradient. The two top panels
of Fig. 12 show the north ($x$) component of the electric field, the two middle panels show the east ($y$) component, and the two
bottom panels show the amplitude. The external, internal, and total DF electric field are shown in the top, third, and fifth panel,
and the total DF, CF, and total electric field in the second, fourth, and sixth panel. Comparison of the external, internal, and
total DF electric field curves reveals that the total DF electric field is sometimes dominated by the external part, sometimes by
the internal part, and sometimes large, oppositely directed peaks in both parts cancel each other out, producing a small total DF
electric field. Comparison of the DF and CF electric field curves show that in the $x$ direction, the CF electric field is directed
in the same way as the DF electric field and has a larger amplitude than the DF electric field, resulting in a total electric field
$x$ component, which is clearly stronger than the DF electric field amplitude. In the $y$ direction, the CF electric field also has a
stronger amplitude than the DF electric field but the CF and DF part are oppositely directed, resulting in a total electric field
that is stronger and oppositely directed to the DF electric field.

### 3.5 Comparison with observations

#### 3.5.1 GIC observations

Recordings of geomagnetically induced currents (GIC) in the Finnish natural gas pipeline were carried out close to the Mäntsälä (MAN) compressor station in southern Finland (60.6 N, 25.2 E) from Nov 1998 to October 2023. The measurements were based on the differential magnetometer method, utilizing two magnetometers: one right above the pipe and the other at the nearby IMAGE station NUR (60.50 N, 24.65 E) (Pulkkinen et al., 2001). The quality of the MAN magnetometer data is variable, but good data are available for the Halloween storm 29–31 October 2003. GIC at MAN can also be estimated from the horizontal geoelectric field using the empirical expression (Pulkkinen et al., 2001; Viljanen et al., 2006)

$$GIC(t) = -70 \text{ A km V}^{-1} \, E_x(t) + 88 \text{ A km V}^{-1} \, E_y(t), \tag{42}$$

which assumes spatially uniform field in the region of the pipeline. In 2003, the magnetometer network around MAN was very sparse and the conductivity model in this region is known to be inaccurate (Marshalko et al., 2023), making direct comparison with the electric field modelled using the SECS method less than ideal. However, we can utilize the simplified version of Section 2.4.

For MAN, the contribution from the nearby magnetometer station NUR is likely to be much larger than that from the other, much more distant, stations. Combined with the linear approximation of Eq. 42, this allows us to estimate the GIC at MAN based on NUR $\frac{\partial \boldsymbol{B}}{\partial t}$ alone

$$GIC(t) = a_x \frac{\partial B_x}{\partial t} + a_y \frac{\partial B_y}{\partial t} + a_z \frac{\partial B_z}{\partial t}. \tag{43}$$

To test this approach, we have used the period from 29 Oct 2003 08:00:00 UT to 31 October 2003 23:59:50 UT to determine the coefficients. The resulting values are provided in Table 1. Observed and modelled GIC for the immediately preceding period from 29 Oct 2003 06:00:00 UT to 07:59:50 UT are shown in Figure 13, and show good agreement in terms of time development and amplitude. The correlation coefficient is $CC = 0.80$ and the coefficient of determination

$$R^2 = 1 - \frac{\sum_i (GIC_i^{obs} - GIC_i^{mod})^2}{\sum_i (GIC_i^{obs} - mean(GIC^{obs}))^2} \tag{44}$$

is $R^2 = 0.64$.

#### 3.5.2 Electric field observations

Electric field observations suitable for comparison with modelling results are rare. However, there is an interval on 11 September 2005 from 05:15:00 UT to 06:15:00 UT with observations (Smirnov et al., 2006) from two sites near the IMAGE station MEK that has been used by Kruglyakov et al. (2023). Kruglyakov et al. (2023) called these sites M02 (63.043740 N, 30.657030 E) and M05 (62.938890 N, 30.993910 E). Similar to MAN, M02 and M05 are located in a region where the magnetometer coverage is far from ideal. Thus, we will again utilize Eq. 27 and approximate the electric field components by fits to $\frac{\partial \boldsymbol{B}}{\partial t}$ from the nearby station MEK (62.77 N, 30.97 E) using Eq. 28–29.

**Table 1.** Coefficients for the linear relation of Eq. 43 between MAN GIC (in A) and NUR $\frac{\partial \boldsymbol{B}}{\partial t}$ (in nT/s) obtained as a fit to the observed GIC from 29 Oct 2003 08:00:00 UT to 31 October 2003 23:59:50 UT. For comparison, the bottom set of coefficients is determined as a fit to the longer period from 29 Oct 2003 00:00:00 UT to 31 October 2003 23:59:50 UT.

| Coefficient | Value | STD error |
|---|---|---|
| $a_x$ | -1.69 | 0.01 |
| $a_y$ | -2.73 | 0.02 |
| $a_z$ | -0.23 | 0.02 |
| $a_x$ | -1.70 | 0.01 |
| $a_y$ | -2.71 | 0.02 |
| $a_z$ | -0.21 | 0.02 |

Because the electric field components provided by Kruglyakov and Marshalko (2023) are in geomagnetic coordinates and the magnetic field observation from IMAGE in geographic, the coefficients in Eq. 28–29 also include rotation of the coordinate system. Geoelectric field observations contain effects due to charge build-up at local small-scale conductivity structures (see, e.g., Kruglyakov et al., 2023). These effects are also included in the coefficients. The coefficients resulting from fitting the period from 05:15:00–05:29:50 UT are provided in Table 2 and the observed and modelled electric field components are shown in Fig. 14. The period used for fitting is shaded. There is good agreement between the observed electric field components and the fits to MEK $\frac{\partial \boldsymbol{B}}{\partial t}$ observations. For the period not used in the fitting, i.e., 05:30:00–06:15:00 UT, the correlation coefficients are 0.91 for M02 $E_x$, 0.95 for M02 $E_y$, 0.95 for M05 $E_x$, and 0.94 for M05 $E_y$. The coefficients of determination are 0.84 for M02 $E_x$, 0.89 for M02 $E_y$, 0.88 for M05 $E_x$, and 0.86 for M05 $E_y$. The good agreement between observations and modelling results here and in Section 3.5.1 support the assumptions of linear dependence between the parameters.

## 4 Discussion

In the previous section we have shown that obtaining the DF part of the geoelectric field from ground-based magnetometer network data using the 2-D SECS method and estimating the CF part from the DF part using Eq. 25–26 can be used to estimate the geoelectric field at the Earth's surface. In cases when the magnetometer network is too sparse for the 2-D SECS method, the same principle of linear relation can be used to estimate the geoelectric field from local $\partial \boldsymbol{B}/\partial t$. In this section we will discuss further applications of the new method.

### 4.1 Maximum amplitudes of the event

The temporal variation in the DF and CF electric field behavior in Fig. 12 is caused by the combination of the spatiotemporally complex external source and the dynamic response of the fixed 3-D ground conductivity. Although induction in the 3-D ground is deterministic, it is unpredictable and difficult to understand, as mentioned in the Introduction. Ideally, we could determine

**Table 2.** Coefficients for the linear relations of Eq. 28–29 between M02 and M05 geomagnetic north ($E_x$) and south ($E_y$) components of the geoelectric field (in $\mathrm{mV/km}$) and MEK geographic $\frac{\partial \boldsymbol{B}}{\partial t}$ (in $\mathrm{nT/s}$) obtained as a fit to the observed geoelectric field on 11 September 2005 from 05:15:00 to 05:29:50 UT.

| Coefficient | Value | STD error |
| --- | --- | --- |
| M02 $q_{xx}$ | -39.85 | 8.74 |
| M02 $q_{xy}$ | 139.11 | 8.82 |
| M02 $q_{xz}$ | -89.54 | 15.13 |
| M02 $q_{yx}$ | -56.73 | 9.28 |
| M02 $q_{yy}$ | 44.27 | 9.37 |
| M02 $q_{yz}$ | -128.25 | 16.07 |
| M05 $q_{xx}$ | 31.64 | 6.59 |
| M05 $q_{xy}$ | -8.34 | 6.65 |
| M05 $q_{xz}$ | 71.20 | 11.41 |
| M05 $q_{yx}$ | -128.24 | 17.67 |
| M05 $q_{yy}$ | 125.37 | 17.83 |
| M05 $q_{yz}$ | -263.88 | 30.59 |

the electric field everywhere in the earth, calculate the telluric current density and charge density, and visualize them in 3-D to understand the complete induction process. However, this is not feasible with present computational resources. Our new method can be used to analyse the spatiotemporal variations of the different geoelectric field contributions: time-variations of ionospheric currents, time-variations of telluric currents, and accumulated charges in the ground. This can help in clarifying
the complicated interaction between the ionosphere and the conducting ground.

As a demonstration, the top part of Table 3 provides the times, locations, and amplitudes of the maximum electric field contributions on the ground in the IMAGE area during a geomagnetic storm between 7 September 2017 23:00:00 UT and 8 September 2017 01:00:00 UT (Dimmock et al., 2019; Juusola et al., 2023a). The DF part of the electric has been calculated using the SECS method with IMAGE magnetic field as input and the CF electric field has been estimated using Eq. 25–26.
The maximum of the external DF electric field took place on 7 Sep 2017 at 23:59:50 UT near stations ABK and AND. At this time, the total DF electric field also had a maximum in the same area, but a slightly larger value had already occurred earlier, at 23:15:30 UT, near station KIR, due to a temporary suppression of the internal part of the DF electric field around that location (not shown). The largest internal DF electric field occurred near station DON on 8 September at 00:30:00 UT. The CF electric field maximum occurred at the same time as the maximum of the external DF field, on 7 Sep 2017 at 23:59:50 UT, at a location
a little southeast of RVK. Due to the very large amplitude of the CF field, this was also the time and location of the total electric field maximum.

**Table 3.** Time, location, and amplitude of the maximum horizontal electric field contributions and time derivative of the horizontal magnetic field ($|\partial \boldsymbol{H}/\partial t|$) contributions in the IMAGE area during a geomagnetic storm between 7 September 2017 23:00:00 UT and 8 September 2017 01:00:00 UT. The DF electric field maxima are given both on the ground (0 km altitude) and in the ionosphere (90 km altitude) and the CF and total electric field peaks as well as $|\partial \boldsymbol{H}/\partial t|$ peaks only on the ground. The CF and total electric field maxima have been determined using the denser ($0.03° \times 0.07°$) PGIEM2G grid and the rest using the sparser ($0.5° \times 1°$) SECS grid.

| Contribution | UT | Geographic latitude, longitude | Amplitude |
|---|---|---|---|
| Ground $|\boldsymbol{E}_{DF,ext}|$ | 7 September 2017 23:59:50 | $68.58°, 19.00°$ | $4100$ mV/km |
| Ground $|\boldsymbol{E}_{DF,int}|$ | 8 September 2017 00:30:00 | $66.00°, 12.00°$ | $3950$ mV/km |
| Ground $|\boldsymbol{E}_{DF}| = |\boldsymbol{E}_{DF,ext} + \boldsymbol{E}_{DF,int}|$ | 7 September 2017 23:15:30 | $67.50°, 21.00°$ | $3753$ mV/km |
| Ground $|\boldsymbol{E}_{CF}|$ | 7 Sep 2017 23:59:50 | $64.54°, 11.91°$ | $70961$ mV/km |
| Ground $|\boldsymbol{E}| = |\boldsymbol{E}_{CF} + \boldsymbol{E}_{DF}|$ | 7 Sep 2017 23:59:50 | $64.54°, 11.91°$ | $72861$ mV/km |
| Ground $|\partial \boldsymbol{H}_{ext}/\partial t|$ | 7 September 2017 23:50:50 | $67.50°, 22.00°$ | $17.52$ nT/s |
| Ground $|\partial \boldsymbol{H}_{int}/\partial t|$ | 8 September 2017 00:03:30 | $66.00°, 27.00°$ | $26.96$ nT/s |
| Ground $|\partial \boldsymbol{H}_{tot}/\partial t|$ | 7 September 2017 23:50:50 | $68.00°, 24.00°$ | $38.65$ nT/s |
| Ionospheric $|\boldsymbol{E}_{DF,ext}|$ | 7 September 2017 23:15:30 | $67.50°, 24.00°$ | $6128$ mV/km |
| Ionospheric $|\boldsymbol{E}_{DF,int}|$ | 8 September 2017 00:29:50 | $66.00°, 12.00°$ | $2897$ mV/km |
| Ionospheric $|\boldsymbol{E}_{DF}| = |\boldsymbol{E}_{DF,ext} + \boldsymbol{E}_{DF,int}|$ | 7 September 2017 23:15:30 | $67.50°, 24.00°$ | $5628$ mV/km |

The total electric field maximum value exceeding 72 V/km may sound high compared to the 20 V/km value cited by Pulkkinen et al. (2012) as an extreme 1-in-100 year geoelectric field event, or to the 8.5 V/km value cited by Lanabere et al. (2024) as an extreme for Sweden. However, both Pulkkinen et al. (2012) and Lanabere et al. (2024) obtained their values using 1-D modelling, whereas our results are based on 3-D modelling. Because 1-D modelling of the electric field does not consider charges, it produces an estimate for the DF electric field. Our maximum DF electric field value of 3753 mV/km is clearly below the extreme values cited by Pulkkinen et al. (2012) and Lanabere et al. (2024). In our modelling, the largest contribution to the maximum electric field comes from the CF electric field, highlighting the importance of 3-D effects when estimating extreme values. We are currently preparing a study on the estimation of the 3-D geoelectric field during extreme geomagnetic storms, and another study on a detailed comparison of 3-D and 1-D geoelectric field modelling.

The middle rows of Table 3 provides the times, locations, and amplitudes of the maximum time derivative of the horizontal magnetic field

$$\left|\frac{\partial \boldsymbol{H}(t)}{\partial t}\right| = \frac{\sqrt{[B_x(t) - B_x(t - \Delta t)]^2 + [B_y(t) - B_y(t - \Delta t)]^2}}{\Delta t} \tag{45}$$

on the ground, which is often used as a proxy for the geoelectric field (Viljanen, 1998; Viljanen et al., 2001). This proxy is based on an approximation of Faraday's law (Eq. 8) at the Earth's surface

$$\frac{\partial B_\theta}{\partial t} = \frac{1}{r}\frac{\partial(rE_{DF,\phi})}{\partial r} = \frac{E_{DF,\phi}}{r} + \frac{\partial E_{DF,\phi}}{\partial r} \approx \frac{E_{DF,\phi}}{R_E} \tag{46}$$

$$\frac{\partial B_\phi}{\partial t} = -\frac{1}{r}\frac{\partial(rE_{DF,\theta})}{\partial r} = -\frac{E_{DF,\theta}}{r} - \frac{\partial E_{DF,\theta}}{\partial r} \approx -\frac{E_{DF,\theta}}{R_E} \tag{47}$$

where the first half of the equations is based on the results of Section 2.2 that at the Earth's surface $E_{DF,r} = 0$. The times and locations of the peak external, internal or total DF electric field and $|\partial \boldsymbol{H}/\partial t|$ values in Table 3 do not match. This demonstrates the complexity of geomagnetic induction as mentioned in Introduction. It also indicates that the drivers of the most intense geoelectric field peaks may not be exactly the same as the drivers of rapid geomagnetic variations (Juusola et al., 2023a).

## 4.2 Directionality of the geoelectric field

Because of the assumed linear dependence of the CF electric field on the DF electric field, it is possible to determine the direction of the driving DF field for which the electric field amplitude at a given location maximizes

$$E_x = k_{xx}E_{DF,x} + k_{xy}E_{DF,y} + E_{DF,x} \tag{48}$$

$$E_y = k_{yx}E_{DF,x} + k_{yy}E_{DF,y} + E_{DF,y} \tag{49}$$

$$E_{DF,x} = |\boldsymbol{E}_{DF}|\cos\alpha \tag{50}$$

$$E_{DF,y} = |\boldsymbol{E}_{DF}|\sin\alpha \tag{51}$$

$$|\boldsymbol{E}| = |\boldsymbol{E}_{DF}|\sqrt{(k_{xx}\cos\alpha + k_{xy}\sin\alpha + \cos\alpha)^2 + (k_{yx}\cos\alpha + k_{yy}\sin\alpha + \sin\alpha)^2} \tag{52}$$

Figure 15 shows maps of the maximum (15a) and minimum $|\boldsymbol{E}|/|\boldsymbol{E}_{DF}|$ (15b), and the angle $\alpha$ of maximum $|\boldsymbol{E}|/|\boldsymbol{E}_{DF}|$ (15c). The value of $|\boldsymbol{E}|/|\boldsymbol{E}_{DF}|$ is the same for $\pm\boldsymbol{E}_{DF}$, which is why there is repetition in the colorbar of the bottom panel. In some areas the maximum $|\boldsymbol{E}|/|\boldsymbol{E}_{DF}|$ is greater than one, indicating that the CF field enhances the DF field, and in some areas that ratio is smaller than one, indicating that the CF field always weakens the DF field. The absolute maximum value $|\boldsymbol{E}|/|\boldsymbol{E}_{DF}| = 40.66$ in the area takes place at $64.48^o$ and $11.77^o$ longitude, and occurs when the DF electric field angle is $\alpha = -52^o$, i.e., the field is directed perpendicular to the nearby coastline, where there are large conductance gradients. The condition that the derivative of Eq. 52 with respect to $\alpha$ is zero is repeated at intervals of $\pi/2$, indicating that the minimum $|\boldsymbol{E}|/|\boldsymbol{E}_{DF}|$ occurs when the DF field is perpendicular to the direction of the maximum enhancement. In most areas the minimum $|\boldsymbol{E}|/|\boldsymbol{E}_{DF}|$ is close to one, i.e., the CF part is close to zero and the DF part determines the electric field amplitude. Fig. 15c emphasizes the active role of the passive Earth in generating geoelectric field peaks.

The above discussion on the dependence of the geoelectric field amplitude on its direction is related to the standard magnetotelluric concept of electric field polarization, typically described using polarization ellipses (Love et al., 2022). The importance of electric field polarization in GIC studies has been highlighted in several papers (Cordell et al., 2021; Love et al., 2019, 2022; Malone-Leigh et al., 2024; Murphy et al., 2021). Exploring the similarities and differences of the polarization as described by polarization ellipses and our method could be yet another interesting topic for a future study.

## 4.3 DF electric field in the ionosphere

DF electric field in the ionosphere is typically ignored although in some dynamical situations inductive effects are not negligible and the ionospheric electric field is not a pure CF field, but has a significant DF part (Vanhamäki et al., 2007; Madelaire et al., 2024). The method used to estimate the DF electric field is valid at and above the Earth's surface. Hence, it can be used to estimate the DF electric field due to telluric and DF horizontal ionospheric currents in the ionosphere as well. The resulting DF electric field is not the total DF electric field above the ionospheric horizontal current sheet, because the contribution from the CF horizontal ionospheric currents and field-aligned currents is missing. Below and at the ionospheric horizontal current sheet, this part of the DF electric field is zero because the combined magnetic field from the horizontal CF currents and (radial) field-aligned currents is zero (Fukushima, 1976; Amm, 1997). Moreover, Vanhamäki et al. (2007) argue that the DF electric field due to the CF horizontal and field-aligned currents should be very small.

Figure 16 shows the external (16a), internal (16b), and total (16c) DF electric field in the ionosphere at 90 $\mathrm{km}$ altitude on 7 September 2017. Similar to the ground (Fig. 3), the external and internal part of the DF electric field are more or less oppositely directed, but whereas on the ground the amplitudes of the external and internal parts were almost equal, in the ionosphere the external part has a clearly stronger amplitude than the internal part. The internal part is also clearly smoother in the ionosphere than on the ground. Nonetheless, the internal part significantly modifies both the pattern and amplitude of the resulting total DF electric field, indicating that ground induction should be included when ionospheric induction is considered. The maximum amplitudes of the total DF electric field, indicated below the scale arrows in Fig. 16, have similar values as the commonly observed ionospheric situations modelled by Vanhamäki et al. (2007).

Vanhamäki et al. (2005) used the 1-D complex image method (CIM) to estimate that the electric field caused by the Earth's induction is relatively small (at most 400 $\mathrm{mV/km}$) and smooth at the ionospheric altitude. This is not in agreement with our result, and indicates that the simple 1-D CIM modelling may not be sufficient for the task.

The bottom part of Table 3 shows the times, locations, and amplitudes of the maximum ionospheric DF electric field contributions in the IMAGE area during a geomagnetic storm between 7 September 2017 23:00:00 UT and 8 September 2017 01:00:00 UT. The largest external and total DF electric field both occurred at the same time and location, on 7 September 2017 at 23:15:30 UT. This was the same time when the DF total electric field maximum occurred on the ground as well. The internal DF electric field had its maximum almost at the same time both in the ionosphere and on the ground.

The method for deriving the CF elecric field from known conductance and vertical current distributions derived in Section 2.3 can also be applied to the ionosphere. In that case, the ground conductance $\Sigma$ is replaced by Pedersen conductance $\Sigma_P$ and terms for the Hall conductance need to be added

$$\boldsymbol{J} = \Sigma_P \boldsymbol{E} + \Sigma_H \boldsymbol{E} \times \hat{\boldsymbol{e}_r} \tag{53}$$

$$\nabla \cdot \boldsymbol{J} = \nabla \Sigma_P \cdot (\boldsymbol{E}_{CF} + \boldsymbol{E}_{DF}) + \Sigma_P \nabla \cdot \boldsymbol{E}_{CF} + \nabla \Sigma_H \cdot [(\boldsymbol{E}_{CF} + \boldsymbol{E}_{DF}) \times \hat{\boldsymbol{e}_r}] + \Sigma_H (\nabla \times \boldsymbol{E}_{DF})_r \tag{54}$$

$$(\nabla \times \boldsymbol{J})_r = [\nabla \Sigma_P \times (\boldsymbol{E}_{CF} + \boldsymbol{E}_{DF})]_r + \Sigma_P (\nabla \times \boldsymbol{E}_{DF})_r + [\nabla \Sigma_H \times ((\boldsymbol{E}_{CF} + \boldsymbol{E}_{DF}) \times \hat{\boldsymbol{e}_r})]_r - \Sigma_H (\nabla \cdot \boldsymbol{E}_{CF}). \tag{55}$$

Eq. 53 is the Ohm's law in a thin-sheet ionosphere, Eq. 54 is the ionospheric equivalent of Eq. 20, and Eq. 55 is a similar expression for the curl of the horizontal current. If $\boldsymbol{E}_{DF}$ is estimated to be insignificant compared to $\boldsymbol{E}_{CF}$, Eq. 54 becomes the traditional problem of the electrostatic ionosphere of global magnetosphere-ionosphere simulations, where $\boldsymbol{E}_{CF}$ is solved from known conductance and vertical current distributions. Although $\boldsymbol{E}_{DF}$ is typically ignored, it would be possible to include it in the calculation, by first using Ohm's law to derive an expression for $I_{DF}(t)$ as a function of the SECS amplitudes $Q_{CF}(t)$, $I_{CF}(t)$ and $I_{DF}(t - \Delta t)$ and inserting this into Eq. 54 to obtain an equation for $Q_{CF}(t)$ in terms of the known $I_{CF}(t)$ and $I_{DF}(t - \Delta t)$. This approach resembles the inductive ionosphere solver presented by Vanhamäki (2011). Ground-based magnetometer networks can be used to determine $(\nabla \times \boldsymbol{J})_r$ and $\boldsymbol{E}_{DF}$, but obtaining $\Sigma_P$ and $\Sigma_H$ is challenging. If they could be estimated, for example from all-sky camera images, Eq. 55 would yield $\boldsymbol{E}_{CF}$. This approach resembles otherwise the corresponding solver suggested by Vanhamäki (2011), except that $\boldsymbol{E}_{DF}$ would be directly obtained from ground-based magnetometer data and, thus, no time-integration would be needed.

A thorough study on the significance of the internal part of the DF electric field on the total ionospheric DF electric field and on the significance of the ionospheric DF electric field to the ionospheric total electric field is a topic for further study. The results may be of interest for global simulations, which typically ignore induction (e.g., Mukhopadhyay et al., 2022; Sorathia et al., 2023; Ganse et al., 2025).

## 5 Conclusions

We have developed a new method for estimating various contributions to the 3-D geoelectric field at the Earth's surface. The surface electric field consists of an external DF electric field due to time-varying ionospheric and magnetospheric currents, an internal DF electric field due to time-varying telluric currents, and a CF electric field due to charge accumulation at ground conductivity gradients.

1. The external part of the DF electric field is calculated from the time derivative of the external part of the observed ground magnetic field using DF 2-D SECSs.

2. The internal part of the DF electric field is calculated from the time derivative of internal part of the observed ground magnetic field using DF 2-D SECSs.

3. The surface CF electric field is calculated from the known surface DF electric field and coefficients that linearly relate the DF electric field to the CF electric field. The coefficiens were obtained from the PGIEM2G model.

4. The calculations are carried out in the time domain, and only two consecutive time steps of the observed magnetic field are needed to compute the surface electric field. This makes the method ideal for near real-time applications. The external part of the DF electric field is valid everywhere, the internal part at and above the Earth's surface, and the CF part at the Earth's surface. A dense magnetometer networks is required for good results.

5. The external and internal parts of the DF electric field are generally oppositely directed and have comparable amplitudes both on the ground and in the ionosphere, indicating that both contributions are significant for the total DF electric field. The largest peaks in the total DF electric field tend to occur when either the external or internal contribution is temporarily suppressed at the location of interest.

6. At a given location, a DF electric field with a given amplitude can result in a total surface electric field amplitude with an orders of magnitude difference depending on the direction of the DF electric field with respect to the locally dominant conductivity gradient structure.

7. Peak amplitudes of the various electric field contributions did not occur at the same time or at the same location as the peak amplitudes of the time derivative of the horizontal magnetic field for our example event. This indicates that analysis of rapid magnetic field variations may not describe all relevant aspects of the electric field behaviour.

8. The linear dependence of the DF electric field on $\partial \boldsymbol{B}/\partial t$ observations on the one hand, and of the CF electric field on the DF electric field on the other hand, makes it possible to estimate the total geoelectric field directly from nearby magnetic field observations. As an example, we have determined coefficients that relate the geolelectric field -driven GIC at MAN to NUR $\partial \boldsymbol{B}/\partial t$.

Analysing the separated contributions from currents (DF electric field) and charges (CF electric field) to the geoelectric field can help in clarifying the complicated interaction between the ionosphere and the conducting ground. As we have shown, the DF field is generally spatially clearly smoother than the CF field. A significant point is that a 3-D ground conductivity leads to charge accumulation at conductivity gradients and to large localized increases of the CF field. Consequently, although the ground conductivity does not depend on time, the response of the ground to external driving is highly dynamical and can produce complicated patterns of the geoelectric field.

*Code and data availability.* IMAGE data (Juusola et al., 2024) are available at https://space.fmi.fi/image. The code for the SECS method is available in Vanhamäki and Juusola (2020). The code used to calculate magnetic coordinates and local times (Laundal et al., 2022) is available at https://apexpy.readthedocs.io/en/latest/. PGIEM2G 3-D EM forward modeling code is developed openly at Gitlab (https://gitlab.com/m.kruglyakov/PGIEM2G) and available under GPLv2. The geomagnetic north and east components of the observed geoelectric field downsampled to 10 seconds (Kruglyakov and Marshalko, 2023) are available at https://zenodo.org/records/8402165. GIC data (Viljanen, 2023) are available at https://space.fmi.fi/gic. The coefficients relating the CF electric field to the DF electric field (Eq. 25–26, Fig. 7–8) are provided as supplementary material.

## Appendix A: Figures for the epoch 7 September 2017 23:52:00 UT

In this appendix we show Figures A1, A2, A3, A4, and A5. They correspond to Fig. 3, 5, 6, 10, and 16, respectively, but illustrate the epoch 7 September 2017 23:52:00 UT instead of the epoch 7 September 2017 23:16:00 UT.

*Author contributions.* LJ implemented the method and prepared the manuscript, HV carried out the PGIEM2G electric field separation into CF and DF parts, and EM provided the PGIEM2G data. MK is the author of PGIEM2G and provided expert advice on geoelectric field modelling. AV provided expert advice on the physics of geomagnetic induction. All co-authors participated in writing the manuscript.

*Competing interests.* The authors declare that they have no conflict of interest.

*Acknowledgements.* We thank the institutes who maintain the IMAGE Magnetometer Array: Tromsø Geophysical Observatory of UiT the Arctic University of Norway (Norway), Finnish Meteorological Institute (Finland), Institute of Geophysics Polish Academy of Sciences (Poland), GFZ German Research Centre for Geosciences (Germany), Geological Survey of Sweden (Sweden), Swedish Institute of Space Physics (Sweden), Sodankylä Geophysical Observatory of the University of Oulu (Finland), Polar Geophysical Institute (Russia), DTU
Technical University of Denmark (Denmark), and Science Institute of the University of Iceland (Iceland). The provisioning of data from AAL, GOT, HAS, NRA, VXJ, FKP, ROE, BFE, BOR, HOV, SCO, KUL, and NAQ is supported by the ESA contracts number 4000128139/19/D/CT as well as 4000138064/22/D/KS. GIC data were recorded in collaboration with Gasum Oy. This research was supported by the Academy of Finland project no. 339329 and no. 354521. MK was supported by the New Zealand Ministry of Business, Innovation, Employment through Endeavour Fund Research Programme contract UOOX2002.

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

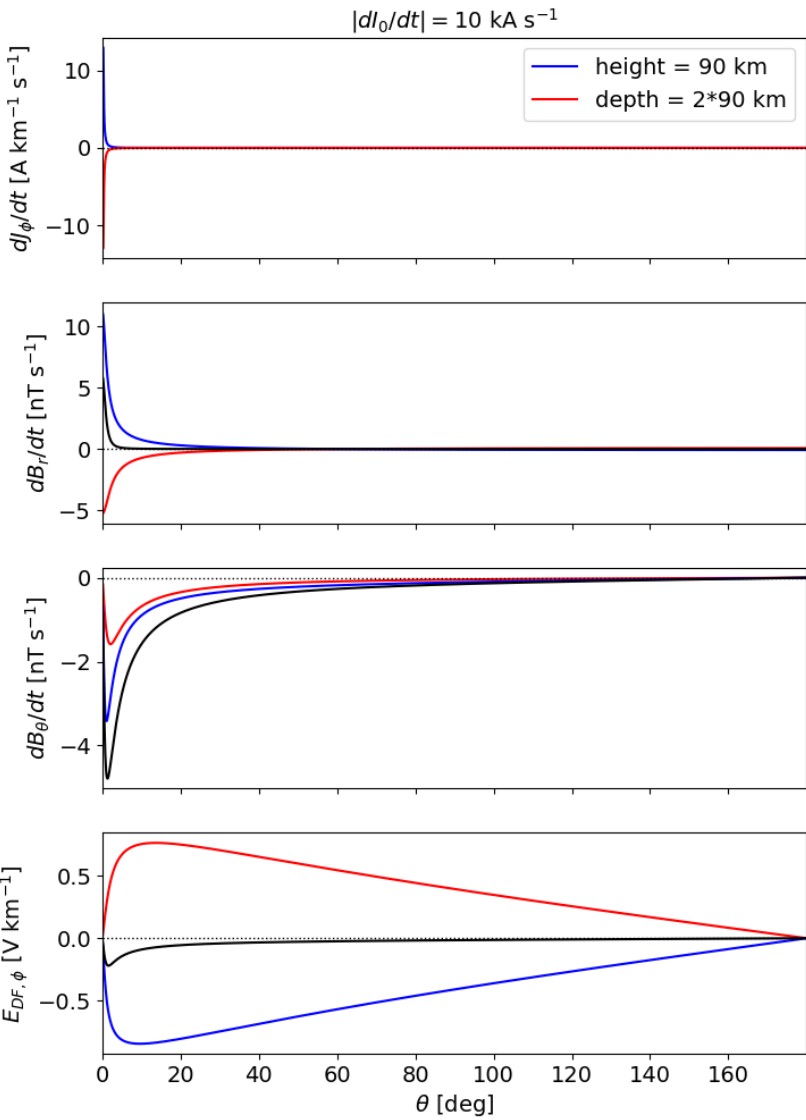

**Figure 2.** Co-latitude profiles of $\partial J_\phi/\partial t$, ground $\partial B_r/\partial t$, ground $\partial B_\theta/\partial t$, and ground $E_{DF,\phi}$ for two different cases: $J_\phi$ of a DF 2-D SECS at 90 km altitude (blue curves, external contribution) and oppositely directed $J_\phi$ at $2\times90$ km depth (red curves, internal contribution). Both have the same rate of change of the SECS amplitude $|\partial I_{DF}/\partial t| = 10$ kA/s. The sum of the external and internal $\partial B_r/\partial t$, $\partial B_\theta/\partial t$, and $E_{DF,\phi}$ on the ground are shown by the black curves.

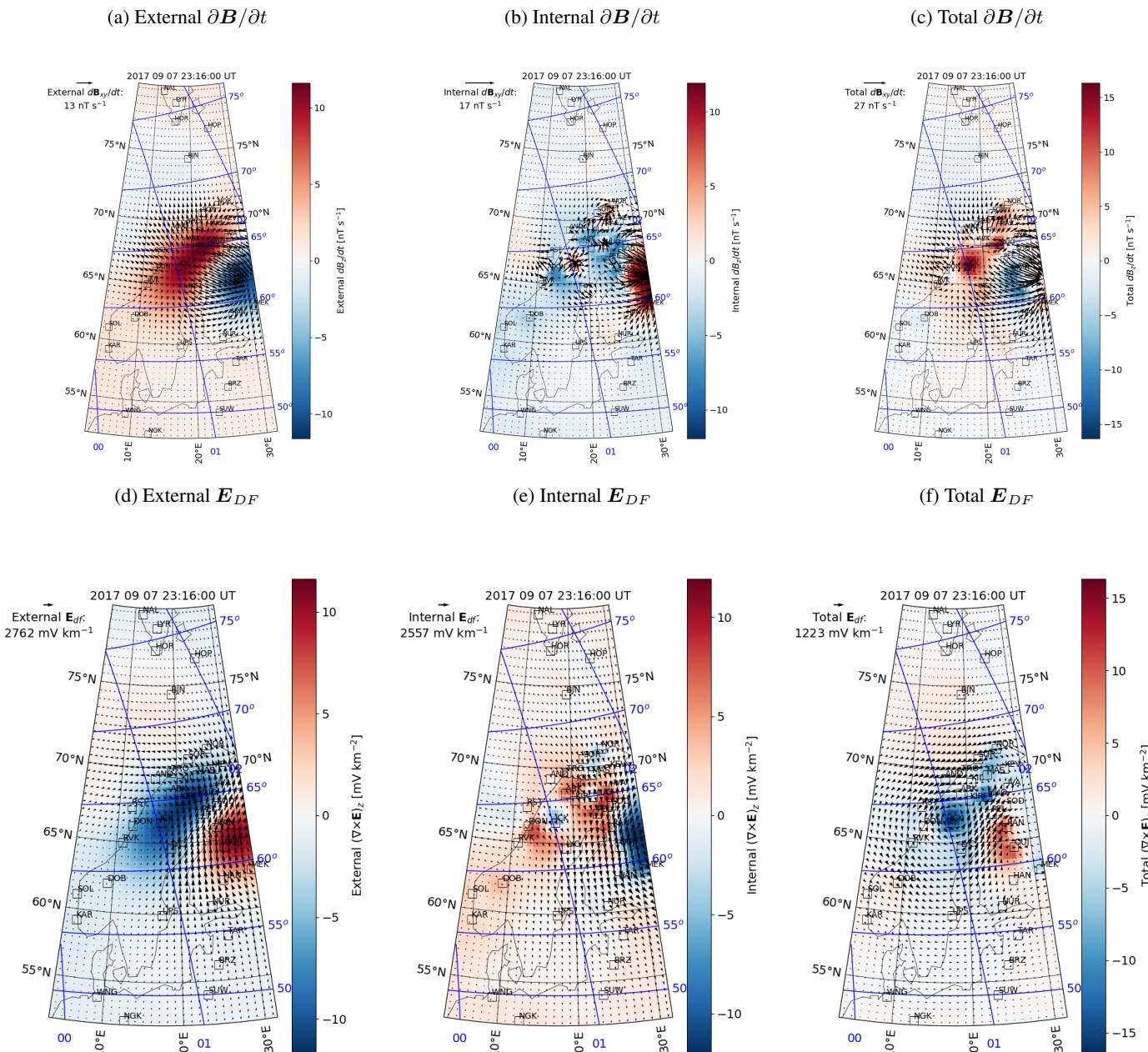

(a) External $\partial\boldsymbol{B}/\partial t$     (b) Internal $\partial\boldsymbol{B}/\partial t$     (c) Total $\partial\boldsymbol{B}/\partial t$

(d) External $\boldsymbol{E}_{DF}$     (e) Internal $\boldsymbol{E}_{DF}$     (f) Total $\boldsymbol{E}_{DF}$

**Figure 3.** External (left), internal (middle), and total (right) time derivative of the magnetic field ($\partial \boldsymbol{B}/\partial t$, top) and DF electric field ($\boldsymbol{E}_{DF}$, bottom) on the ground on 7 Sep 2017 at 23:16 UT. The horizontal component is shown by vectors in all plots. In the top row, the vertical component of $\partial \boldsymbol{B}/\partial t$ is shown by color, and in the bottom row, the curl of $\boldsymbol{E}_{DF}$. Note that the color and arrow length scales vary between panels. Locations of the IMAGE magnetometer stations used in the analysis are indicated by the black squares. Apex coordinates (Richmond, 1995; Emmert et al., 2010; Laundal et al., 2022) are indicated with the blue grid. The north, east, and down components $(B_x, B_y, B_z)$ used in the plots correspond to $(-B_\theta, B_\phi, -B_r)$.

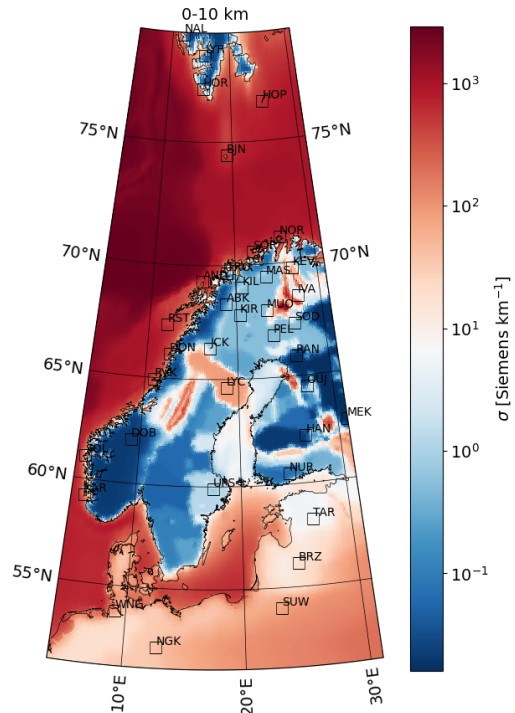

**Figure 4.** Conductivity in the 0–10 km layer (i.e., the top layer, which is 10 km thick) according to the SMAP model (Korja et al., 2002). The colorbar maximum and minimum value correspond to the largest and smallest conductivity value, respectively, placing the white color of the diverging color map in the middle of these values.

(a) $\nabla \cdot \boldsymbol{E}$ (CF SECS, $j_r = 0$)  (b) $|\mathbf{E}_{CF}|$ (CF SECS, $j_r = 0$)  (c) $|\mathbf{E}|$ (CF SECS, $j_r = 0$)

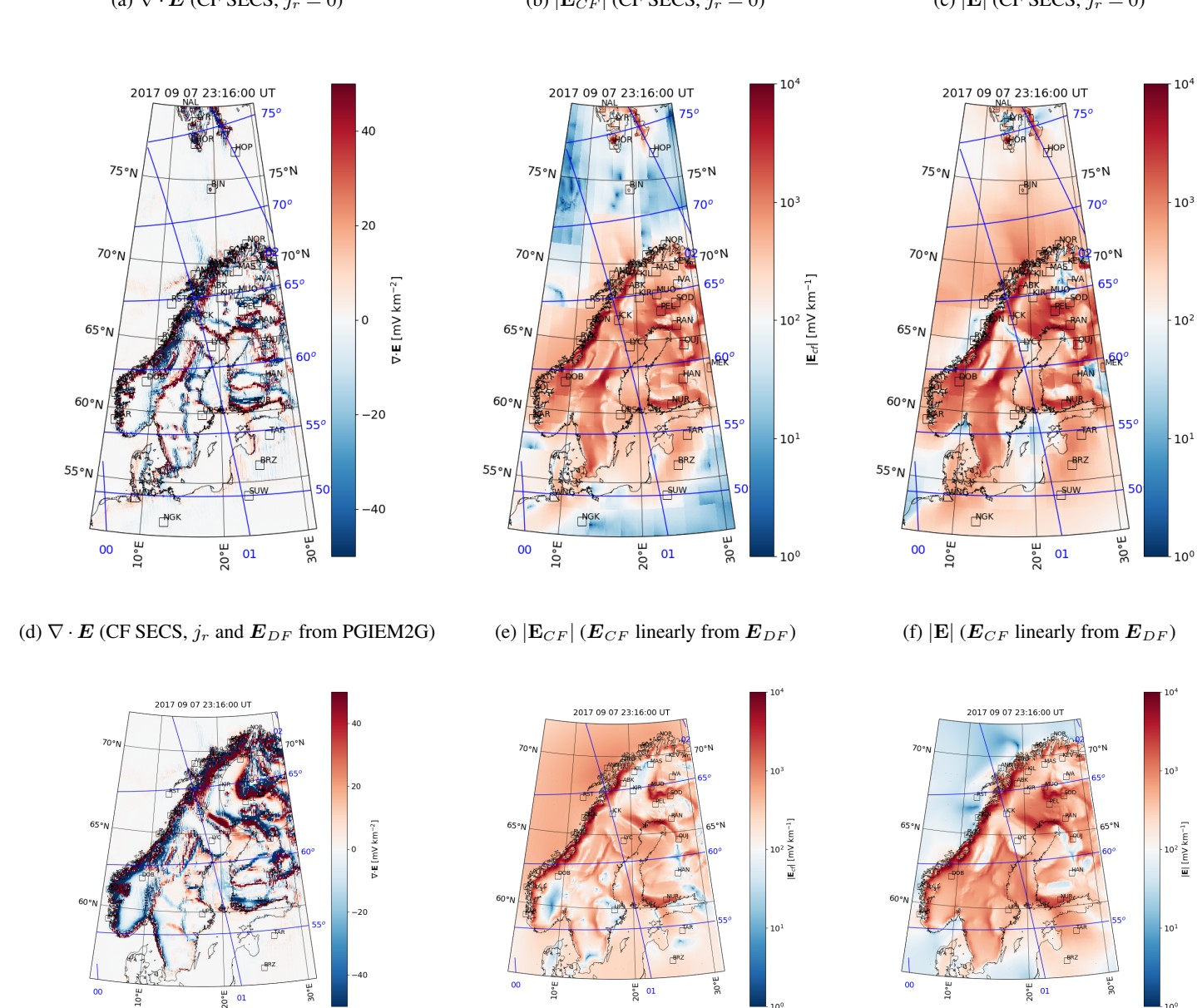

(d) $\nabla \cdot \boldsymbol{E}$ (CF SECS, $j_r$ and $\boldsymbol{E}_{DF}$ from PGIEM2G)  (e) $|\mathbf{E}_{CF}|$ ($\boldsymbol{E}_{CF}$ linearly from $\boldsymbol{E}_{DF}$)  (f) $|\mathbf{E}|$ ($\boldsymbol{E}_{CF}$ linearly from $\boldsymbol{E}_{DF}$)

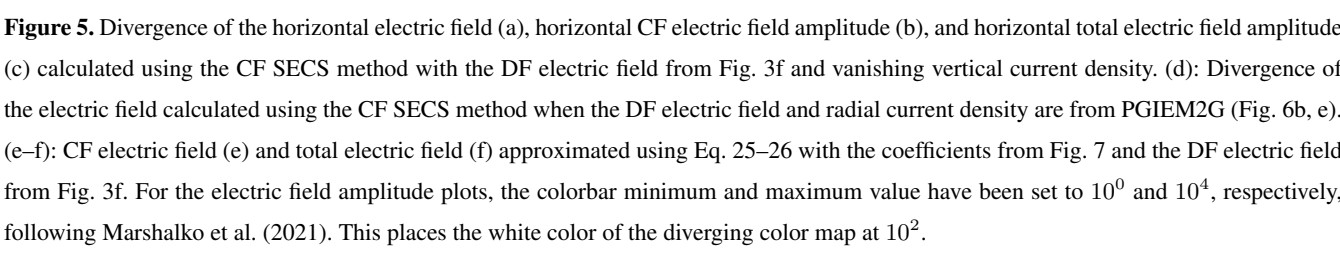

**Figure 5.** Divergence of the horizontal electric field (a), horizontal CF electric field amplitude (b), and horizontal total electric field amplitude (c) calculated using the CF SECS method with the DF electric field from Fig. 3f and vanishing vertical current density. (d): Divergence of the electric field calculated using the CF SECS method when the DF electric field and radial current density are from PGIEM2G (Fig. 6b, e). (e–f): CF electric field (e) and total electric field (f) approximated using Eq. 25–26 with the coefficients from Fig. 7 and the DF electric field from Fig. 3f. For the electric field amplitude plots, the colorbar minimum and maximum value have been set to $10^0$ and $10^4$, respectively, following Marshalko et al. (2021). This places the white color of the diverging color map at $10^2$.

(a) $|\mathbf{E}|$ (PGIEM2G)

(b) $\mathbf{E}_{DF}$ and $(\nabla \times \mathbf{E})_z$ (PGIEM2G)

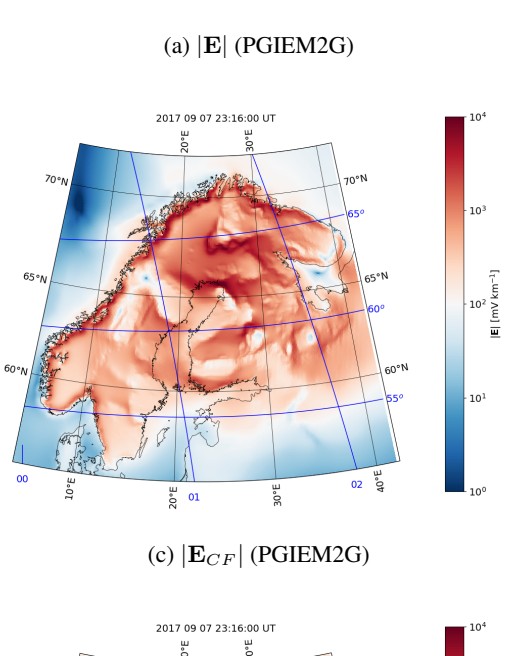

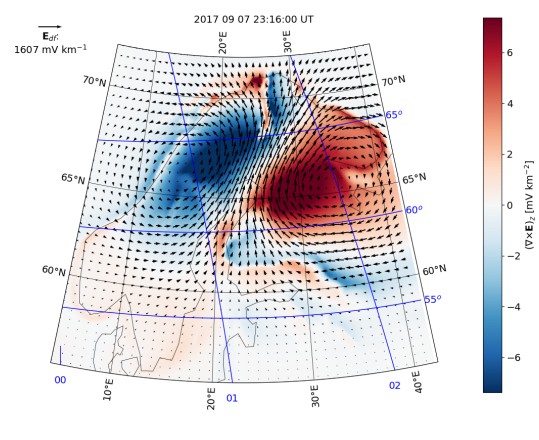

(c) $|\mathbf{E}_{CF}|$ (PGIEM2G)

(d) $\nabla \cdot \mathbf{E}$ (PGIEM2G)

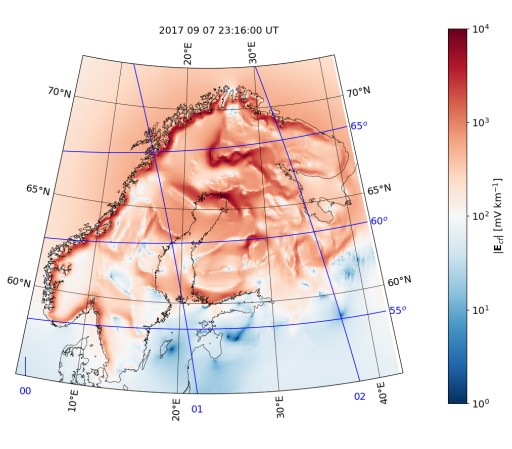

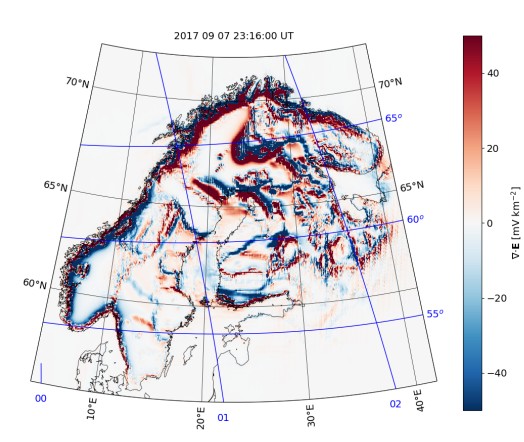

(e) $\nabla \cdot \mathbf{J}/\Sigma$ (PGIEM2G)

(f) $\nabla \cdot \mathbf{E}$ (CF SECS, $j_r$ and $\mathbf{E}_{DF}$ from PGIEM2G)

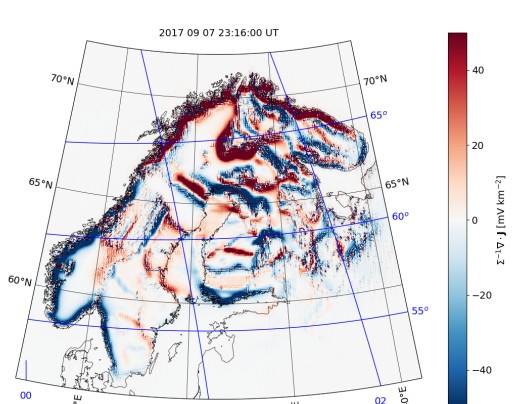

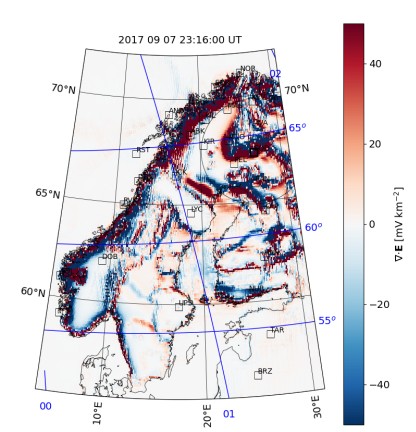

**Figure 6.** Horizontal electric field amplitude modelled using PGIEM2G (Marshalko et al., 2021) (a), DF part of the electric field and its curl (arrows and color in panel b), CF part of the electric field (c), divergence of the electric field (d), and divergence of the horizontal current density divided by conductance (bote). The bottom right panel shows the divergence of the electric field derived using the SECS method when the DF electric field and radial current density are from PGIEM2G.

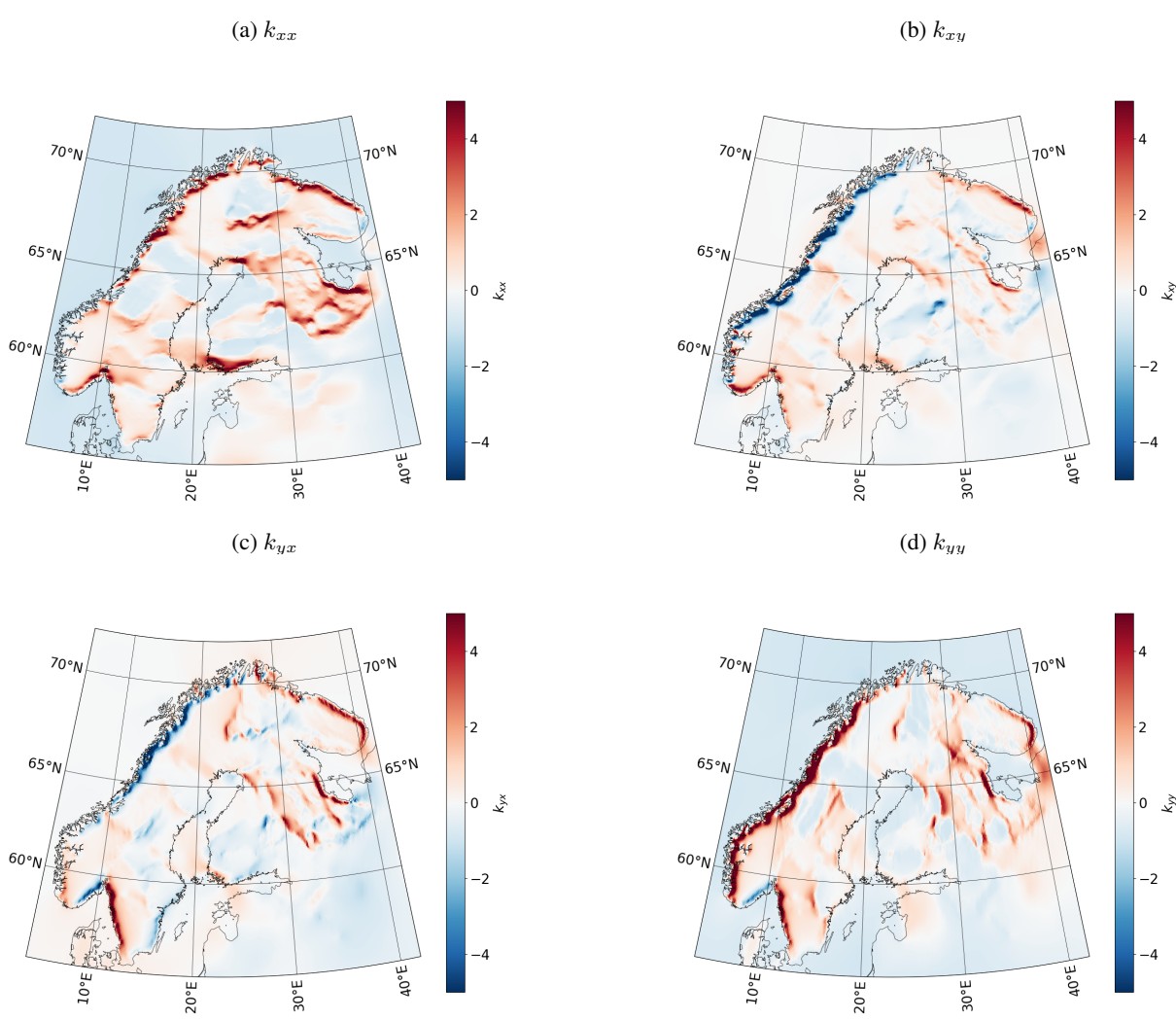

(a) $k_{xx}$

(b) $k_{xy}$

(c) $k_{yx}$

(d) $k_{yy}$

**Figure 7.** Proportionality coefficients (Eq. 25–26) derived using the CF and DF part of PGIEM2G's horizontal surface electric field between 7 September 2017 23:00 UT and 8 September 2017 01:00 UT.

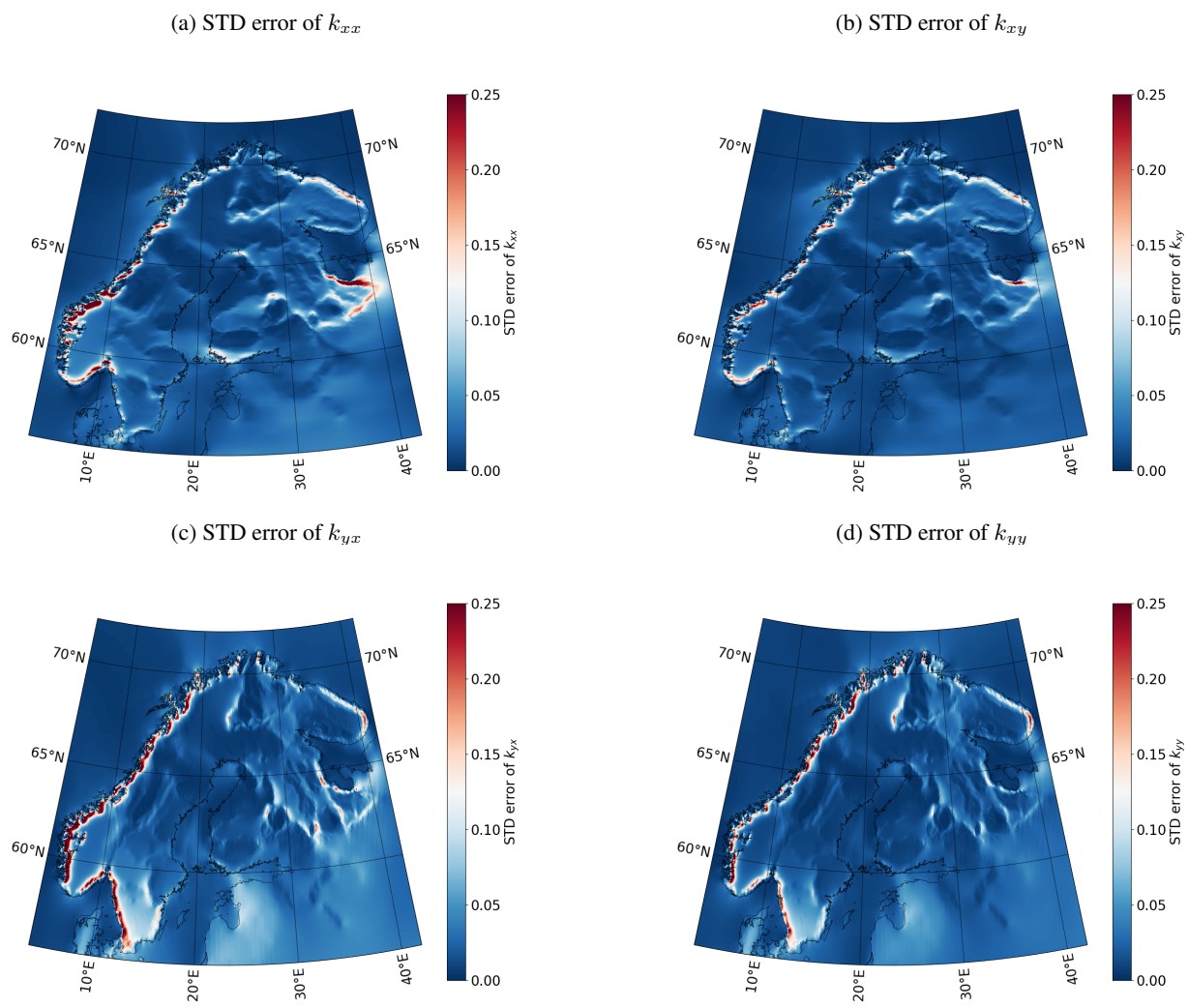

**Figure 8.** The same as Figure 7 except that instead of the coefficients their standard deviation (STD) errors are shown. The colorbar minimum and maximum value have been set to $0$ and $0.25$, respectively, placing the white color of the diverging color map in the middle of these values.

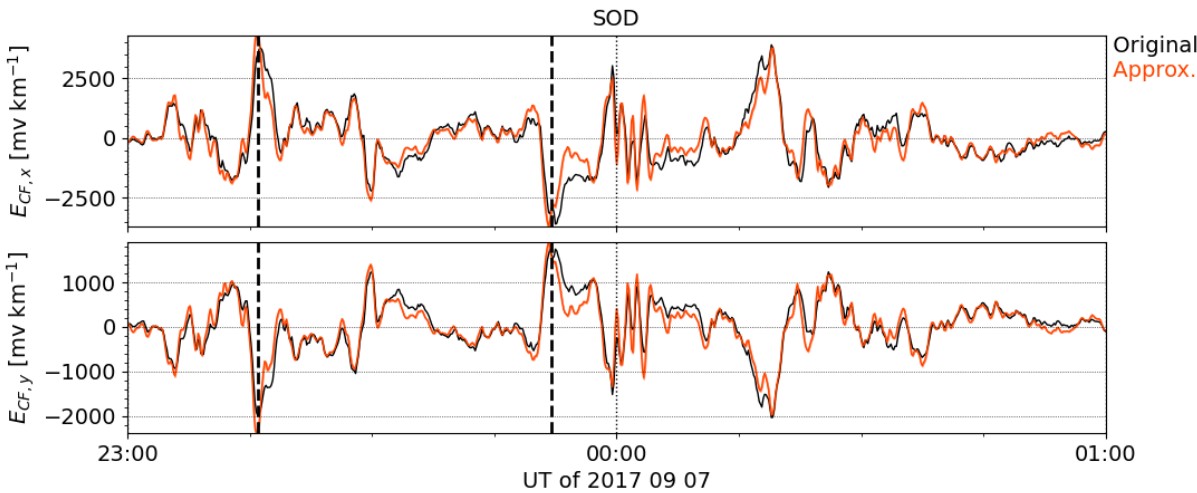

**Figure 9.** CF electric field at SOD between 7 September 2017 23:00 UT and 8 September 2017 01:00 UT. The original PGIEM2G-modelled value is shown in black and an approximation based on Eq. 25–26 is shown in red.

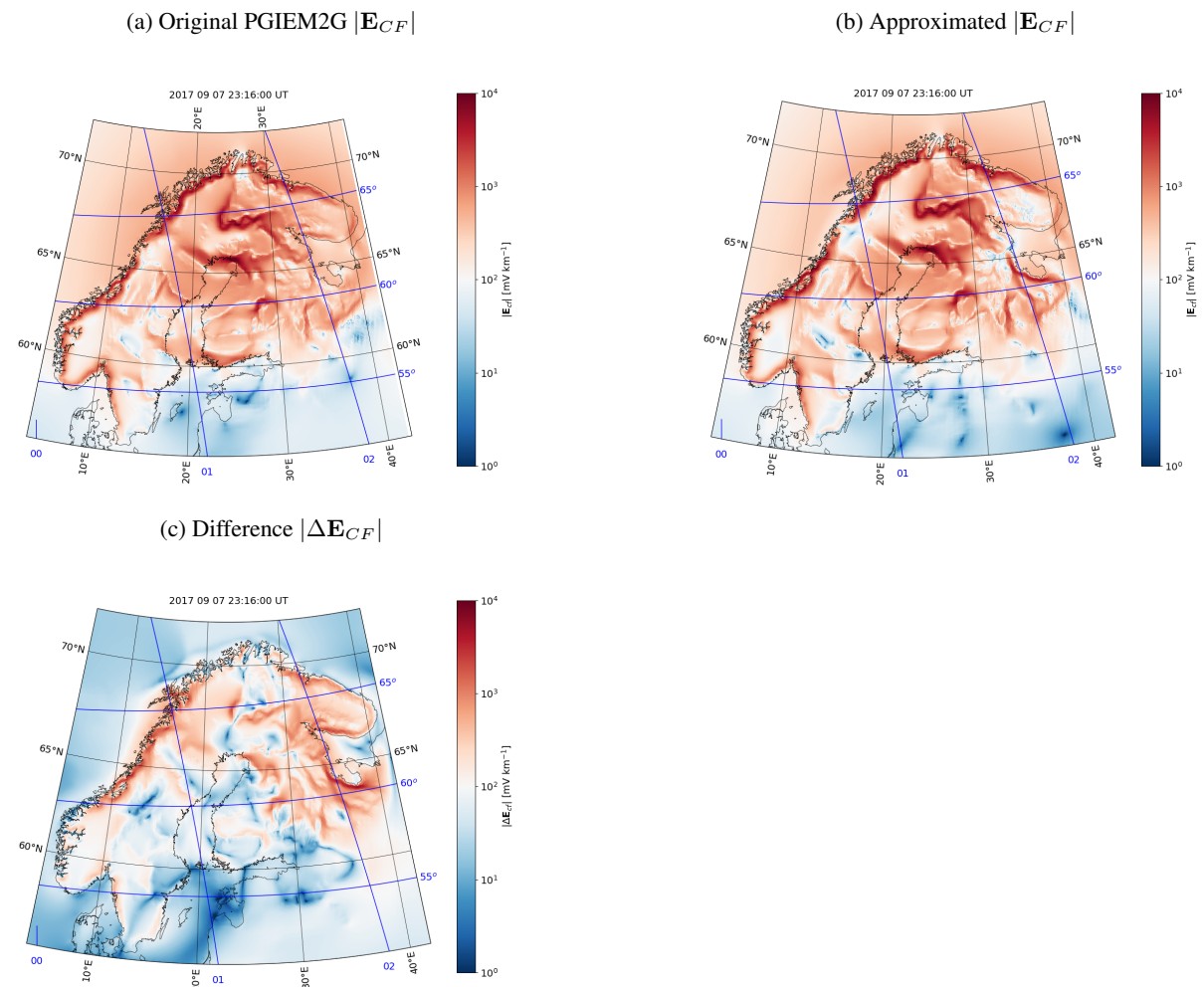

**Figure 10.** CF part of the horizontal electric field modelled by PGIEM2G on 7 Sep 2017 at 23:16:00 UT (a), CF electric field approximated using Eq. 25–26 with the coefficients from Fig. 7 and DF electric field from PGIEM2G (Fig. 6b) (b), and the difference between the original and approximated CF field (c).

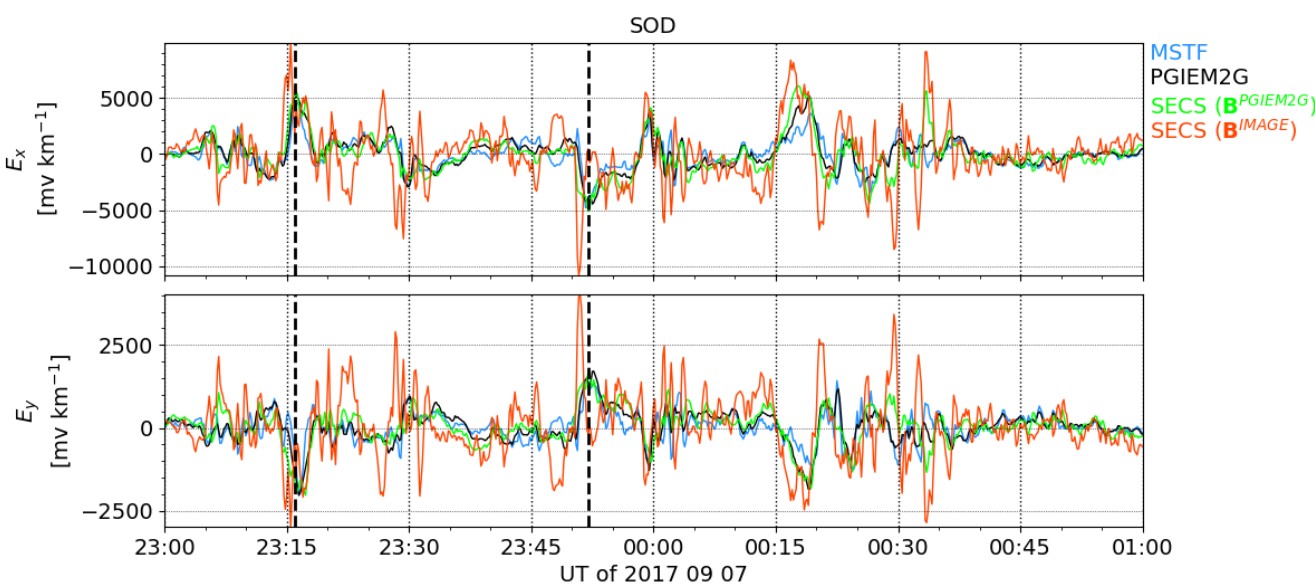

**Figure 11.** $E_x$ and $E_y$ at SOD between 7 September 2017 23:00 UT and 8 September 2017 01:00 UT. PGIEM2G modelling is shown in black and SECS (DF electric field from SECS, CF electric field from Eq. 25–26) modelling in red (calculated using the magnetic field from IMAGE) and green (calculated using the magnetic field from PGIEM2G). The multi-site transfer function (MSTF) approach (Kruglyakov et al., 2023) is shown in blue.

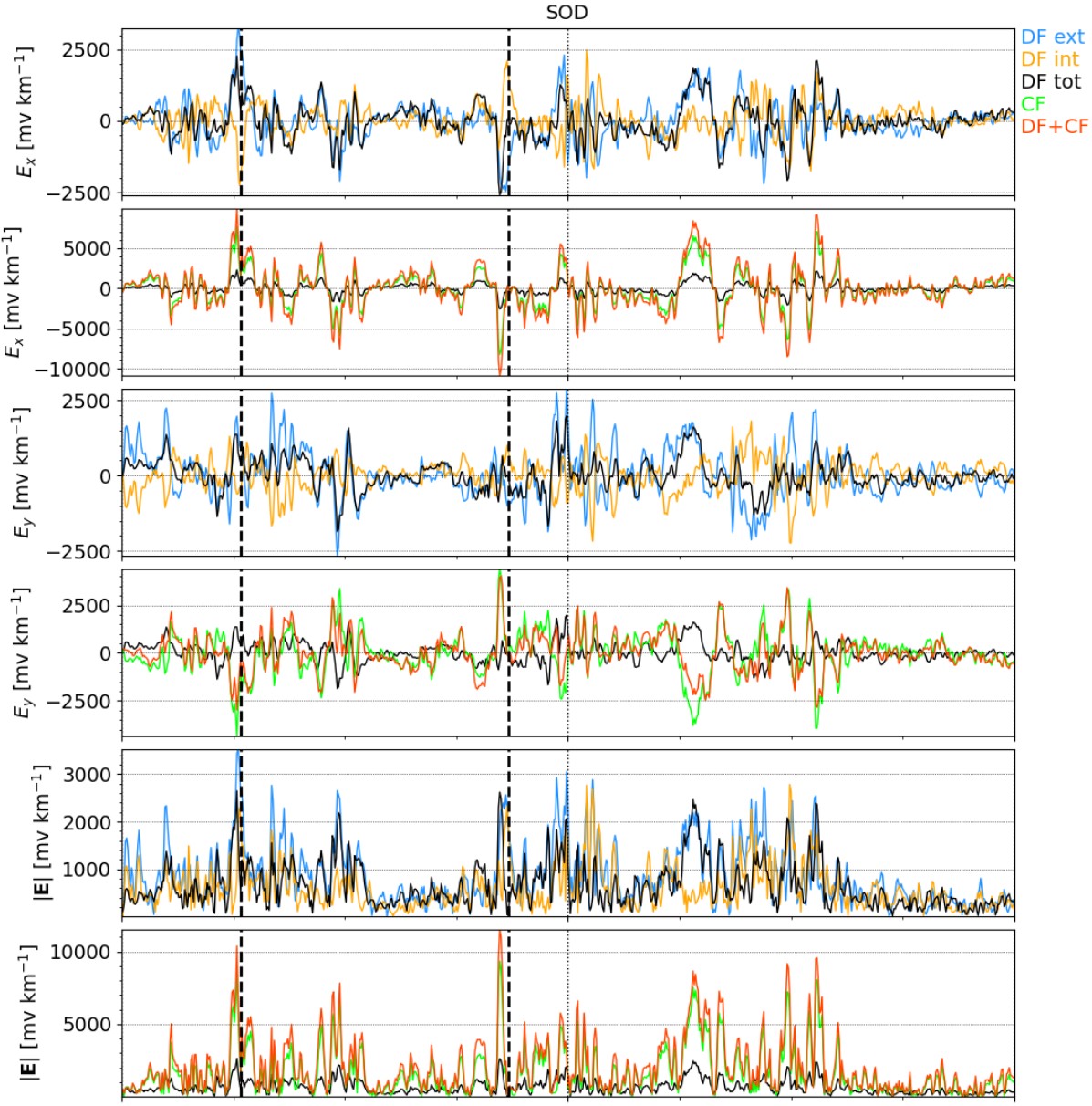

**Figure 12.** Time series of the north ($x$) and east ($y$) components and amplitudes of the external ($\boldsymbol{E}_{DF,ext}$) and internal ($\boldsymbol{E}_{DF,int}$) part of the DF electric field, total ($\boldsymbol{E}_{DF} = \boldsymbol{E}_{DF,ext} + \boldsymbol{E}_{DF,int}$) DF electric field, CF electric field ($\boldsymbol{E}_{CF}$), and total ($\boldsymbol{E}_{DF} + \boldsymbol{E}_{DF}$) electric field at the location of the magnetometer station SOD between 7 September 2017 23:00 UT and 8 September 2017 01:00 UT. The DF electric field has been modelled using the DF SECS method and IMAGE data, and the CF electric field has been obtained using the coefficients from Fig. 7. The vertical dashed lines indicate the epochs 23:16 UT and 23:52 UT displayed in Fig. 3, Fig. 5, Fig. A1, and Fig. A2.

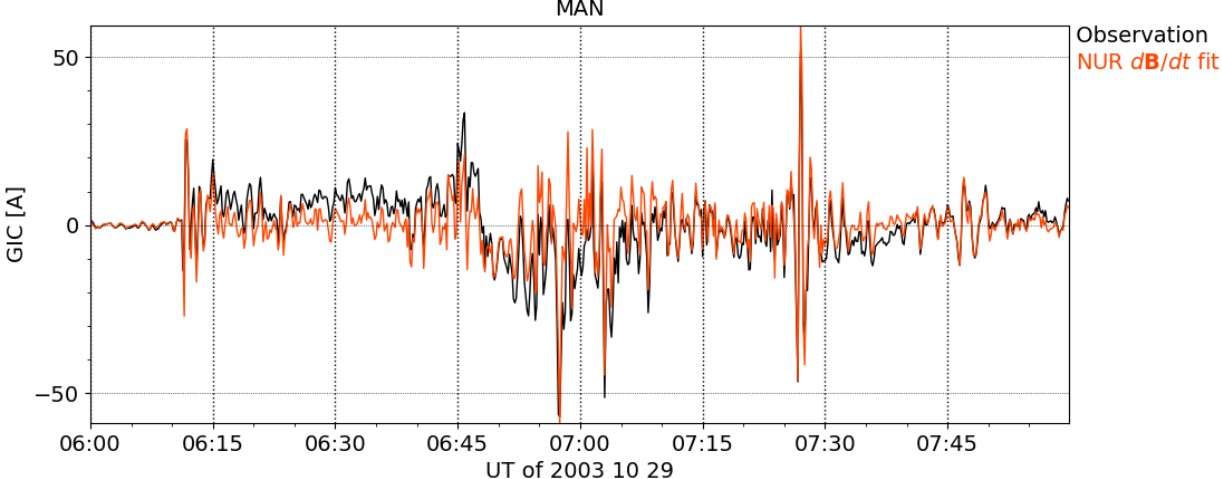

**Figure 13.** GIC observed at MAN on 29 Oct 2003 from 06:00:00 UT to 07:59:50 UT (black) and GIC modelled based on NUR magnetic field observations using Eq. 43 and the coefficients provided in Table 1.

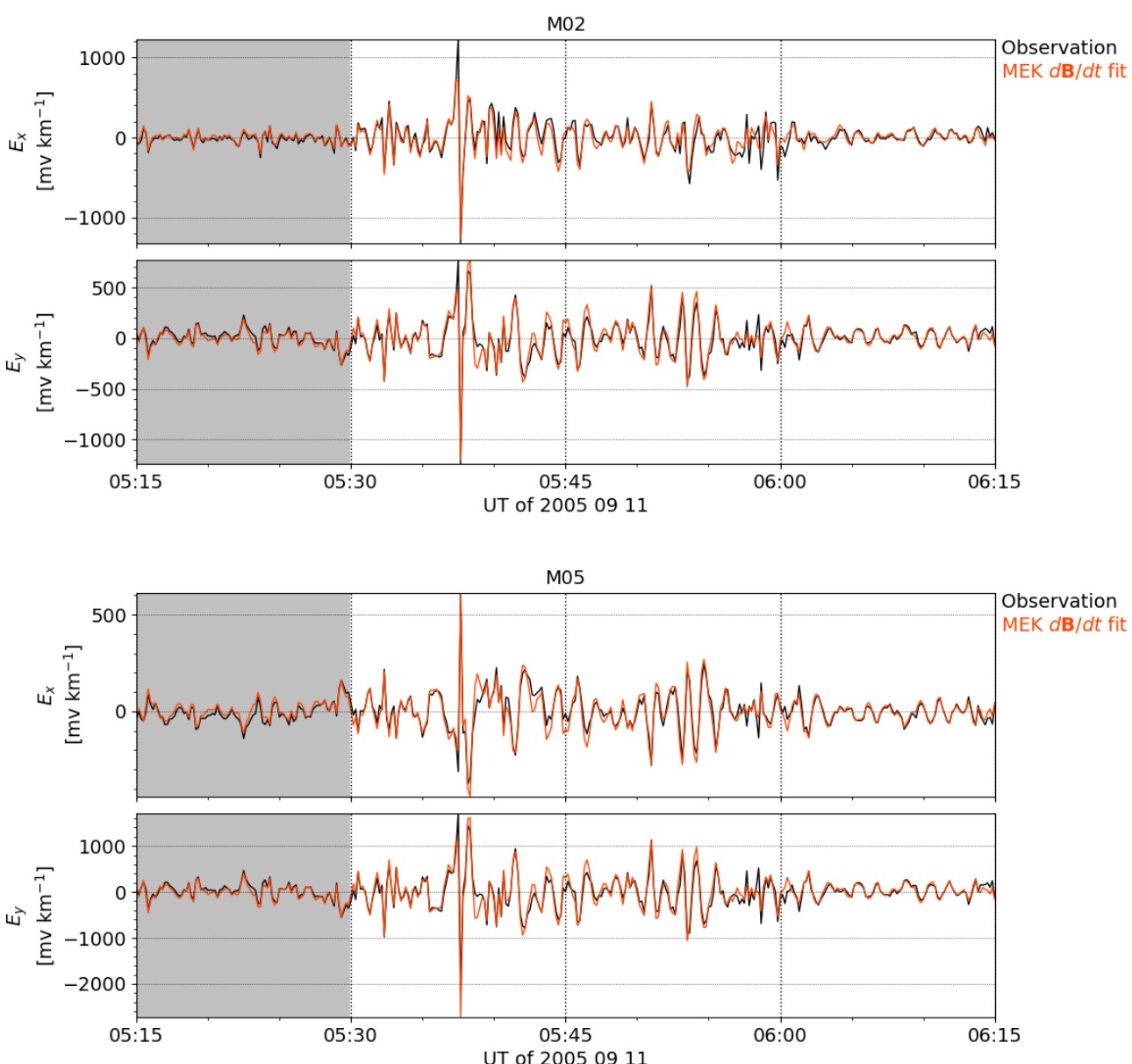

**Figure 14.** Geomagnetic north ($E_x$) and east ($E_y$) components of the geoelectric field observed at the sites M02 (63.043740 N, 30.657030 E) and M05 (62.938890 N, 30.993910 E) on 11 September 2005 from 05:15:00 to 06:15:00 UT (black) and $E_x$ and $E_y$ modelled based on nearby MEK (62.77 N, 30.97 E) geographic magnetic field observations using Eq. 28–29 and the coefficients provided in Table 2. The interval used to determine the coefficients is shaded.

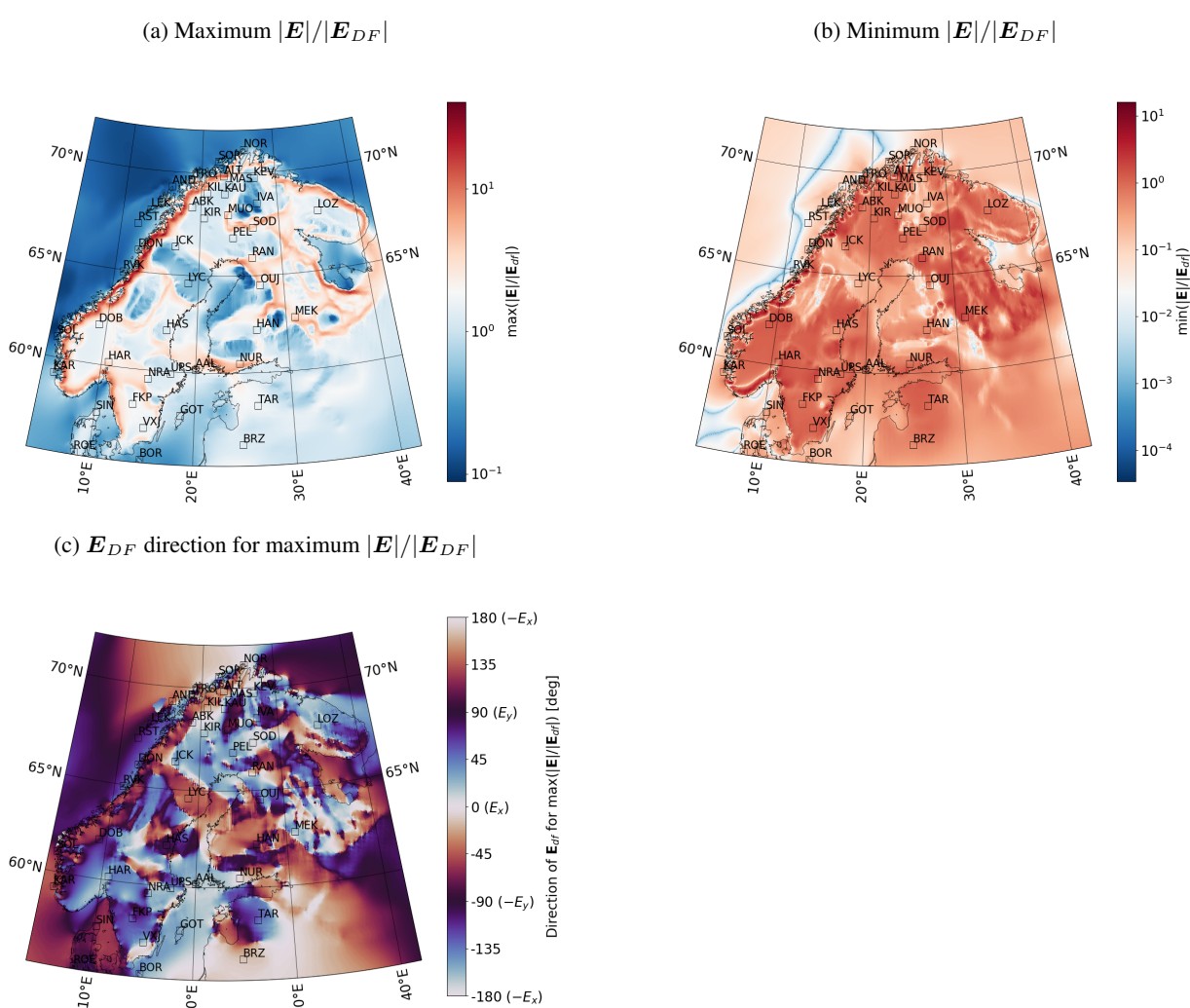

**Figure 15.** Maximum (a) and minimum $|\boldsymbol{E}|/|\boldsymbol{E}_{DF}|$ (b), and DF electric field direction where maximum $|\boldsymbol{E}|/|\boldsymbol{E}_{DF}|$ is reached (c). For the panels in the top row, the colorbar maximum and minimum value correspond to the largest and smallest value of the shown parameter, respectively, placing the white color of the diverging color map in the middle of these values.

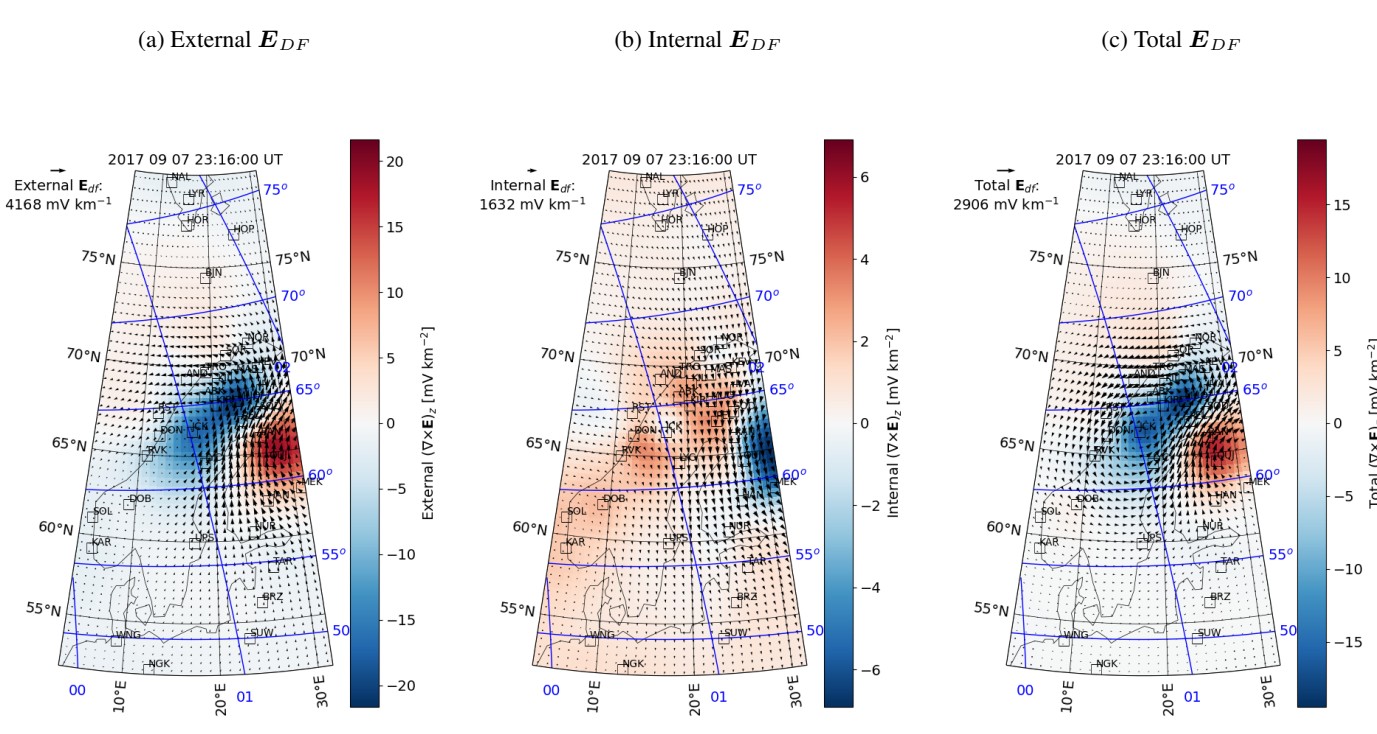

**Figure 16.** External (a), internal (b), and total (c) DF electric field (arrows) and its curl (color) in the ionosphere at 90 km altitude on 7 September 2017 at 23:16:00 UT.

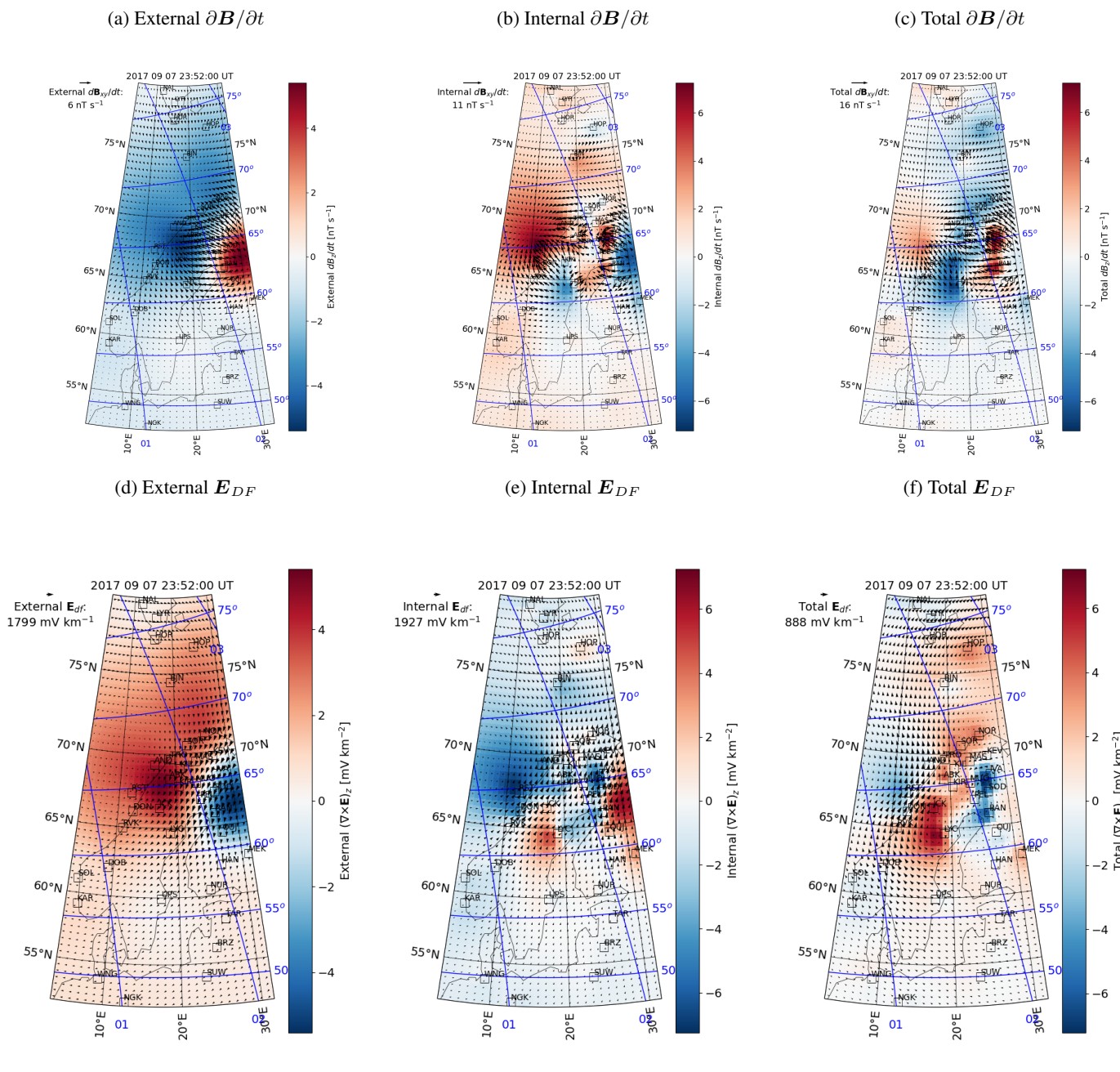

**Figure A1.** The same as Fig. 3 except for 23:52:00 UT instead of 23:16:00 UT.

(a) $\nabla \cdot \boldsymbol{E}$ (CF SECS, $j_r = 0$)  (b) $|\mathbf{E}_{CF}|$ (CF SECS, $j_r = 0$)  (c) $|\mathbf{E}|$ (CF SECS, $j_r = 0$)

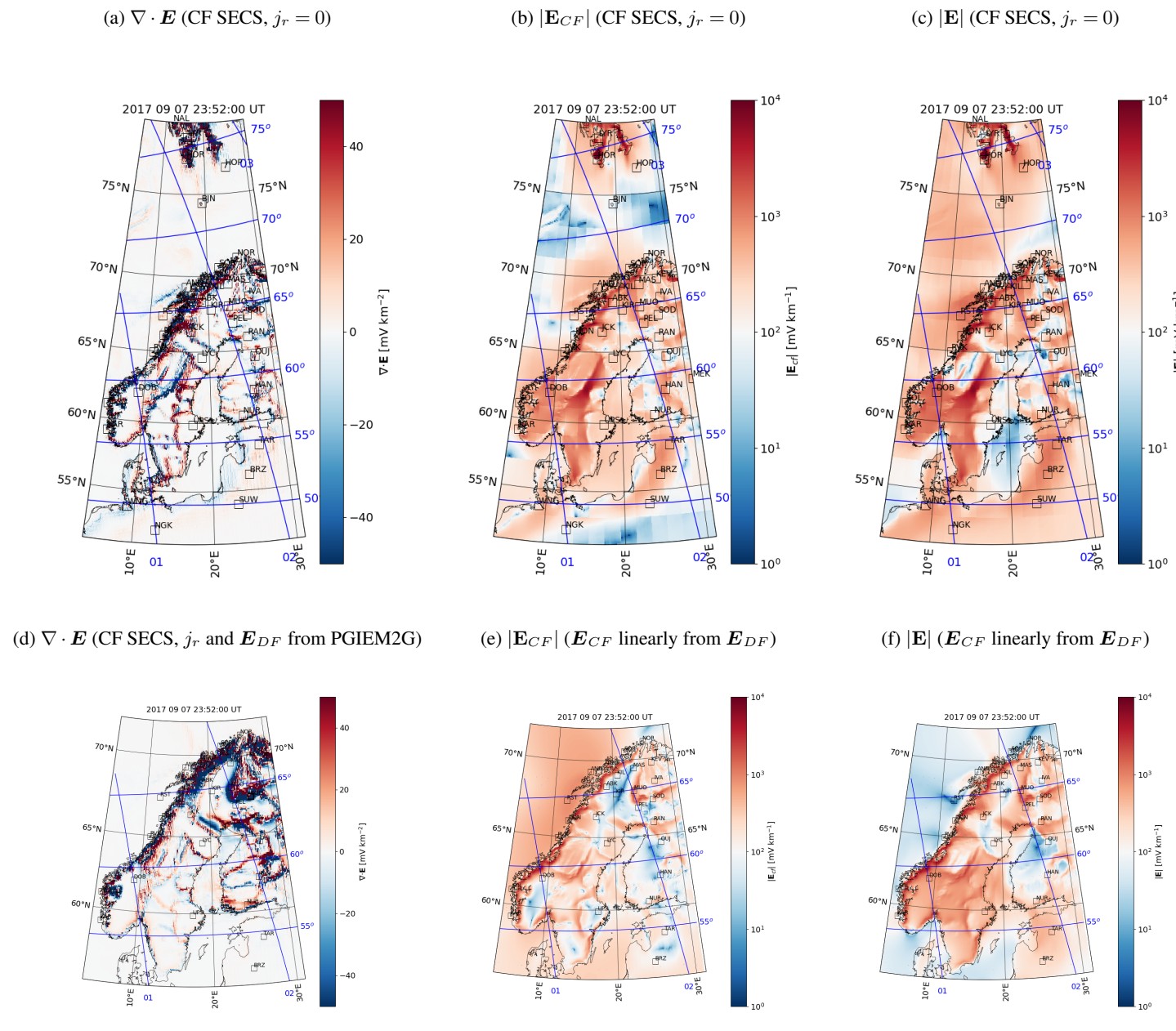

(d) $\nabla \cdot \boldsymbol{E}$ (CF SECS, $j_r$ and $\boldsymbol{E}_{DF}$ from PGIEM2G)  (e) $|\mathbf{E}_{CF}|$ ($\boldsymbol{E}_{CF}$ linearly from $\boldsymbol{E}_{DF}$)  (f) $|\mathbf{E}|$ ($\boldsymbol{E}_{CF}$ linearly from $\boldsymbol{E}_{DF}$)

**Figure A2.** The same as Fig. 5 except for 23:52:00 UT instead of 23:16:00 UT.

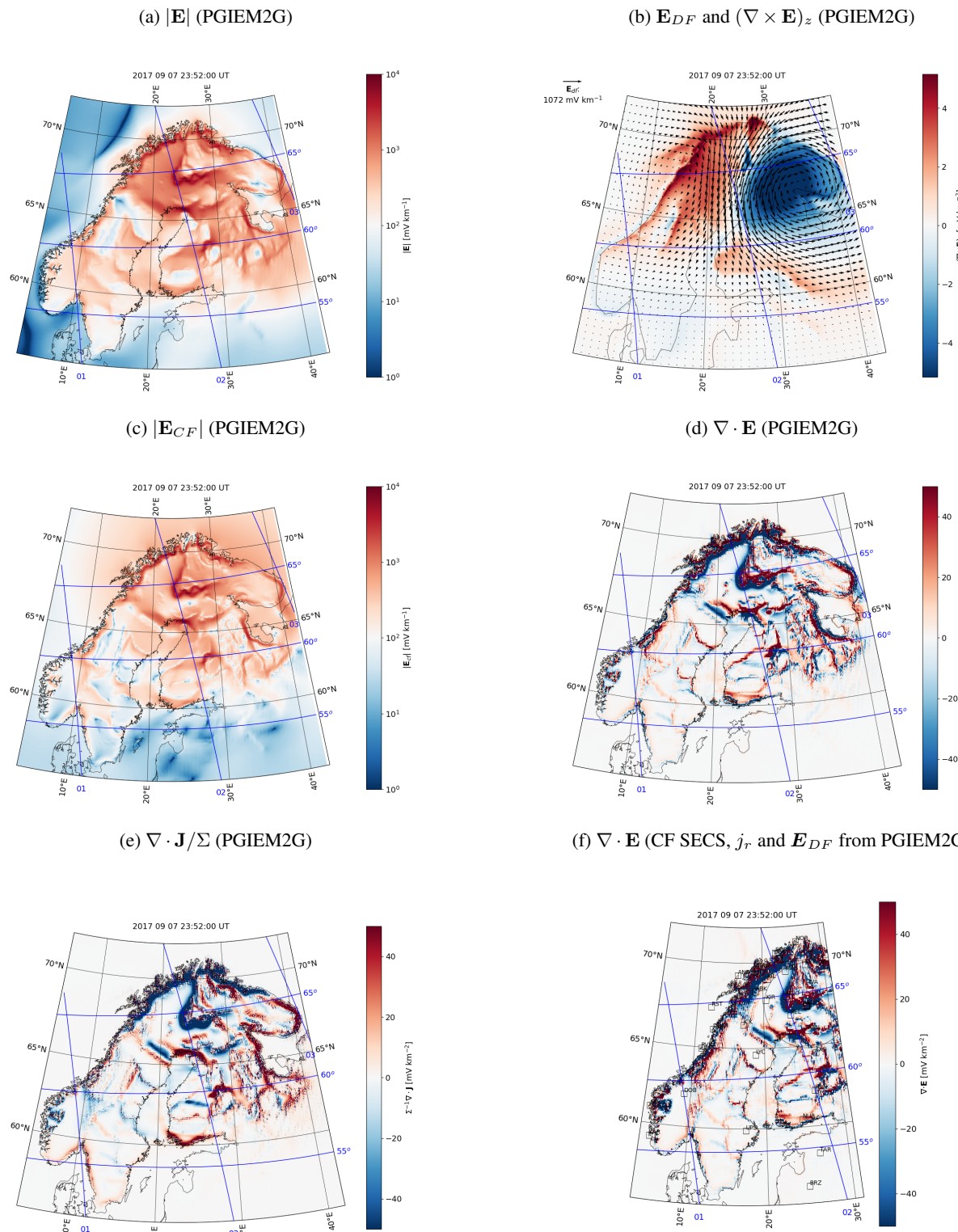

**Figure A3.** The same as Fig. 6 except for 23:52:00 UT instead of 23:16:00 UT.

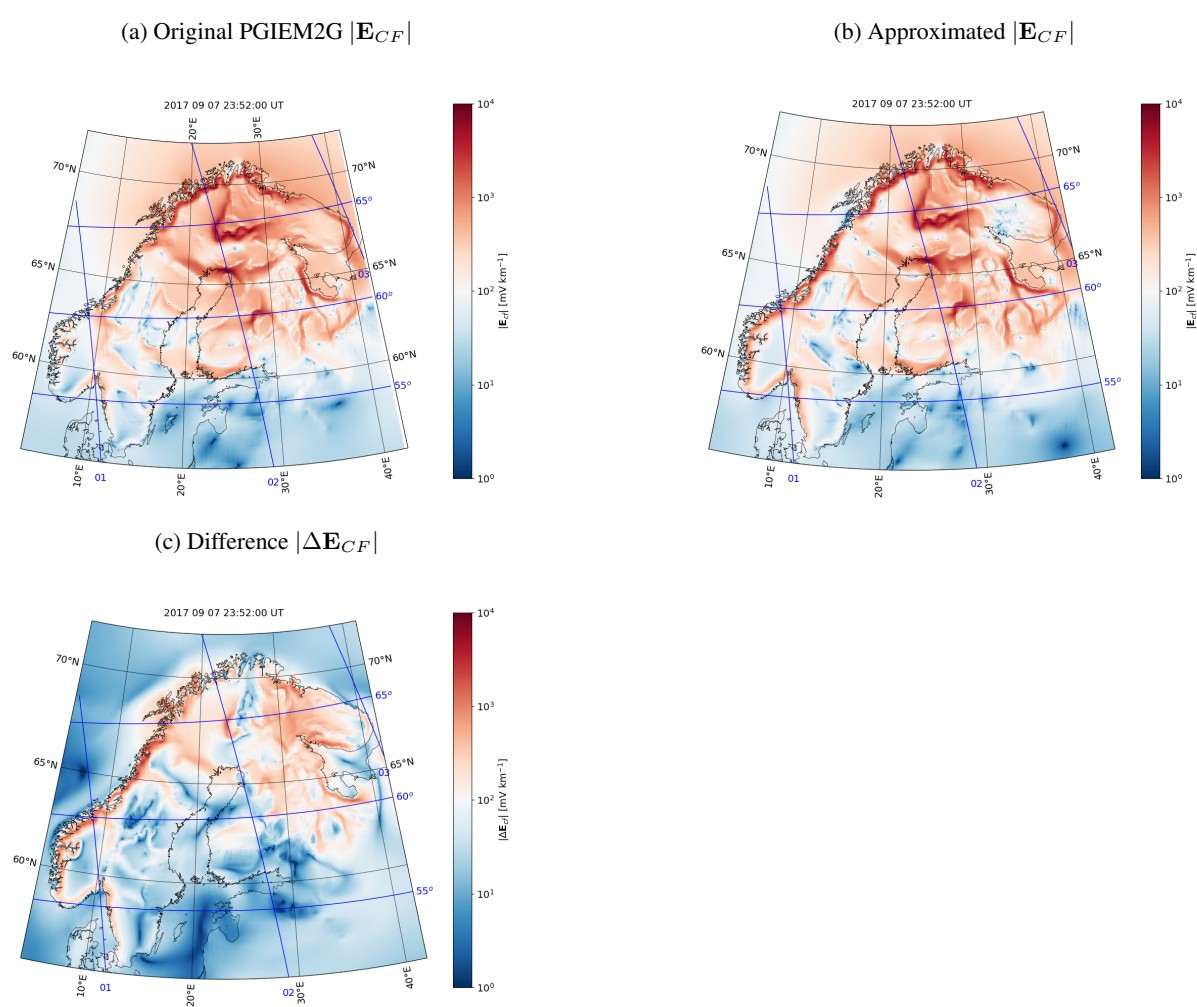

(a) Original PGIEM2G $|\mathbf{E}_{CF}|$

(b) Approximated $|\mathbf{E}_{CF}|$

(c) Difference $|\Delta\mathbf{E}_{CF}|$

**Figure A4.** The same as Fig. 10 except for 23:52:00 UT instead of 23:16:00 UT.

(a) External $\boldsymbol{E}_{DF}$    (b) Internal $\boldsymbol{E}_{DF}$    (c) Total $\boldsymbol{E}_{DF}$

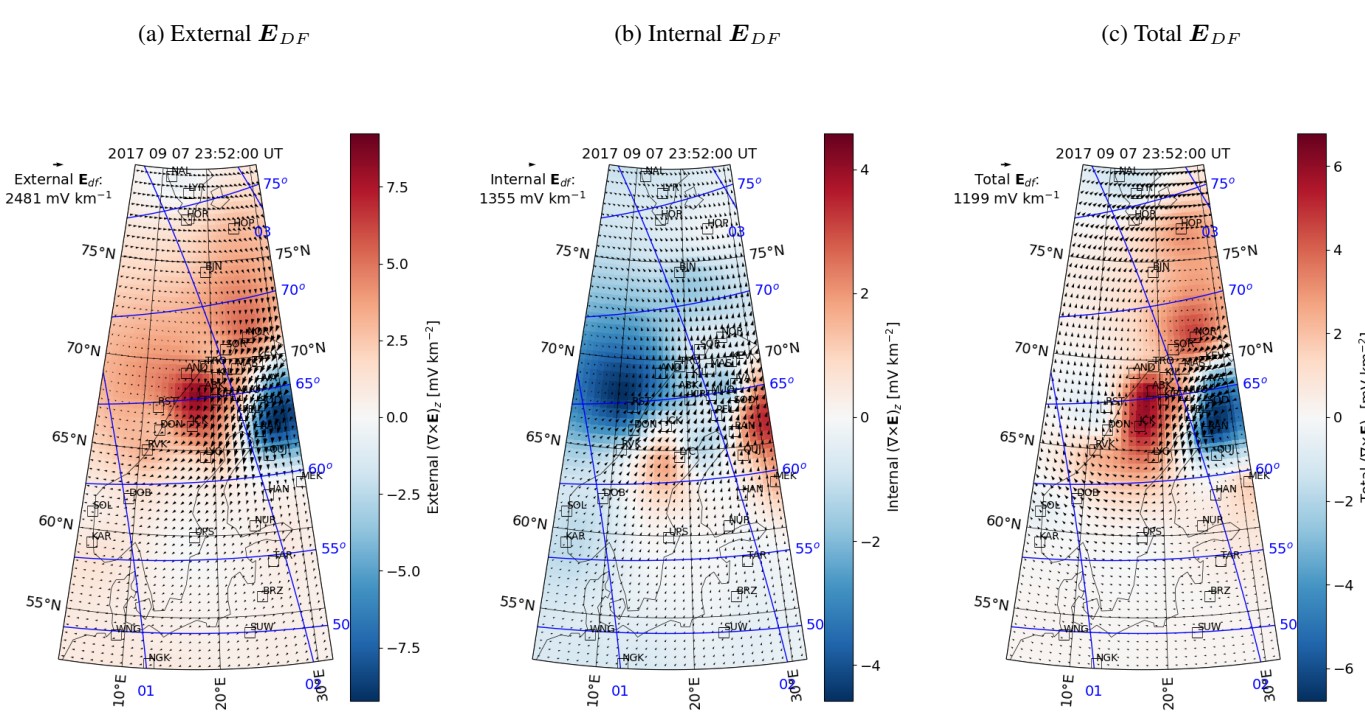

**Figure A5.** The same as Fig. 16 except for 23:52:00 UT instead of 23:16:00 UT.