# Peer review of "Estimation of the 3-D geoelectric field at the Earth's surface using Spherical Elementary Current Systems"

_EGUsphere, 2024_

## Author Comment (AC1)

**Reply to Referee #1**

Liisa Juusola[1], Heikki Vanhamäki[2], Elena Marshalko[1], Mikhail Kruglyakov[3], and Ari Viljanen[1]

[1]Finnish Meteorological Institute, Helsinki, Finland
[2]University of Oulu, Oulu, Finland
[3]University of Otago, Dunedin, New Zealand

**Dear Referee #1,**

thank you for very useful and constructive comments. Please, see below for our point-by-point replies. The original review is written in *black* and our replies in blue.

*Review comments on the manuscript egusphere-2024-2831, entitled: Estimation of the 3-D geoelectric field at the Earth's*

5 *surface using Spherical Elementary Current Systems*

*by Liisa Juusola et al.*

*The authors tried to derive the geoelectric field at the Earth's surface from magnetic field variations measured in the vicinity. The various components of the E-field are estimated with the help of the SECS approach and by using the 3-D induction model PGIEM2G. This approach is applied to IMAGE Magnetometer Network area. Convincing results are obtained in this*

10 *way, which compare quite favorably with GIC measurements in gas pipelines. The computational design of the framework is suitable for running it in near-real time for estimating space weather hazards, resulting from GICs in the Fenno-Scandian region.*

*In spite of these generally positive ratings, the study would gain, when improvements were made in a number of cases.*

*Open issues*

15    1. *One thing, the authors obviously have not taken into account is the effect of prompt penetrating electric fields on the geoelectric field. As shown by Brändlein et al. (2012) doi:10.1029/2012JA018008, the ionospheric Hall current, driven by the prompt penetration field, causes ground-based magnetic signatures, but it does not cause geoelectric fields on the ground. At mid-latitudes significant effects of this process can be observed. I am not aware that anyone has studied this effect at auroral latitudes. This point should be discussed.*

20    Thank you for bringing this to our attention. We suggest to add at line 132:

"This follows from neglecting the displacement current, as is usual in geoelectromagnetism. However, it should be noted that the displacement current may play a role in producing ground-based magnetic field signatures that do not cause a geoelectric field on the ground (Brändlein et al., 2012)."

and at line 151:

25    "Brändlein et al. (2012) discussed a waveguide transmission, where the wave mode on the ground has a non-zero horizontal magnetic field component but a zero horizontal electric field. In the vertical direction such a wave mode is expected to have a zero magnetic and non-zero electric field components. The SECS reconstruction is able to reproduce such a magnetic field as a superposition of the magnetic fields of ionospheric and telluric equivalent currents. Because the vertical magnetic field is zero, $E_{DF}$ would also be zero."

2. *Table 3: Larges ground E-fields are predicted at the end of 7 Sep. 2017 for a location close to the transformer of Namsos. It should be checked if measurements of ground currents are available at that station. In case there are, they should be compared with the predictions. This would make the study much more convincing and relevant for application.*

Unfortunately, the Namsos GIC measurement did not start until 2020. Furthermore, in order to predict the GIC, the network geometry and resistances would be needed. This information is generally classified.

3. *In the Introduction it is mentioned that a second layer is introduces below the Earth's surface. From the following sections it is not clear what this extra layer physically represents. How does it account for lateral conductivity variations?*

This is a good point. We suggest to modify the text at line 78. The original text is

"... a second layer just below the Earth's surface. This..."

and the modified text would be

"... a second layer just below the Earth's surface, to represent the magnetic field of the telluric currents. A ground conductivity model is not needed for the DF electric field calculation, because the conductivity distribution affects the telluric current distribution, and this is reflected in the magnetic field it produces. Using two layers instead of one..."

4. *Another statement is that the radial component of EDF is not required to be zero. What is the effect of that assumption? What does it physically imply? These two latter assumptions are pointed out as important assets of the presented approach. Therefore, they should be better explained to the readers.*

We do not make any requirements about the radial direction of EDF, but show that, because of the geometry of the DF SECS, EDF only has a horizontal component between the ground surface and the ionospheric sheet current. In order to try to make this more clear, we suggest to modify the text at line 132. The original text is

"Consequently, the induced electric field only has a $\phi'$ component,"

and the modified text would be

"Due to the geometry of the DF SECS magnetic field, the corresponding induced electric field only has a $\phi'$ component (see also the vector potential derivation in Amm and Viljanen (1999)),"

Furthermore, we suggest to add at line 151, after the addition suggested in our reply to point 1:

"Although the combination of ionospheric and telluric DF current densities always produces a DF electric field that has a zero vertical component between the ionspheric and telluric equivalent current sheets, this does not necessarily mean that $E_{DF}$ cannot have a vertical component in this region. This issue was investigated in detail by Pirjola and Viljanen (1998). In addition to the parts described by the DF current densities, the 3D current distributions in the ionosphere and in the ground include a part that has a zero magnetic field between the ionosphere and ground surface. However, the corresponding vector potential $A$ may not be zero, although $\nabla \times A$ must be zero. The corresponding DF electric field is a Laplace field that has its sources above the ionosphere and inside the ground. A similar Laplace electric field could also be produced with electric charges in these regions. According to the results by Pirjola and Viljanen (1998), valid up to neglecting the displacement current, any horizontal part of this DF field is cancelled by charges accumulated on the ground surface, leaving only an insignificant vertical component. Thus this part of the induction process does not drive any GIC."

In addition, we will remove some extra $\epsilon_0$ from Eq. 22 and Eq. 24. This is just a typo and does not affect the results.

**References**

Amm, O. and Viljanen, A.: Ionospheric disturbance magnetic field continuation from the ground to ionosphere using spherical elementary current systems, Earth Planets Space, 51, 431–440, https://doi.org/https://dx.doi.org/10.1186/BF03352247, 1999.

Brändlein, D., Lühr, H., and Ritter, O.: Direct penetration of the interplanetary electric field to low geomagnetic latitudes and its effect on magnetotelluric sounding, J. Geophys. Res., 117, A11314, https://doi.org/https://doi.org/10.1029/2012JA018008, 2012.

Pirjola, R. and Viljanen, A.: Complex image method for calculating electric and magnetic fields produced by an auroral electrojet of finite length, Ann. Geophys., 16, 1434–1444, https://doi.org/https://doi.org/10.1007/s00585-998-1434-6, 1998.

70

---

## Author Comment (AC2)

**Reply to Referee #2**

Liisa Juusola[1], Heikki Vanhamäki[2], Elena Marshalko[1], Mikhail Kruglyakov[3], and Ari Viljanen[1]

[1]Finnish Meteorological Institute, Helsinki, Finland
[2]University of Oulu, Oulu, Finland
[3]University of Otago, Dunedin, New Zealand

**Dear Referee #2,**

thank you for very useful and constructive comments. Please, see below for our point-by-point replies. The original review is written in *black* and our replies in blue.

5   *The authors describe a new set of techniques to model the geoelectric field using curl free as well as the divergence free geomagnetic field. They work through a series of simplification of Maxwell's equations to derive the relationships and point out interesting insights into the induced geoelectric field properties. The model does require a good representation of the ground conductivity which can be a limitation for many other locations. Overall this is an excellent contribution to the research area and will be interesting to apply in locations outside the Scandinavian region.*

10   *Minor comments:*

*Abstract: I would not have a citation embedded in the abstract ((Kruglyakov & Kuvshinov, 2018)*
We will remove the citation.

*Line 9: coefficients*
Will be corrected.

15   *Line 17: with orders of magnitude*
Will be corrected.

*Line 24: , a solid understanding*
Will be corrected.

*Line 25: scarce*
20   Will be corrected.

*Line 28: A couple of more linking sentences would be useful. E.g. To achieve an intercomparison of results we ... "do things ..."*
We suggest to add: "We will approach the modelling problem by separating the different contributions to the geoelectric field."

25   *Line 52: surface*
Will be corrected.

*Line 150: You make an excellent point about the induced fields tending to cancel each other out.*

Thank you!

*Line 188: geoelectric*

Will be corrected.

*Line 300: It is not entirely clear at this point that the SMAP model with PGIEM2G is a prerequisite for the modelling to work to compute CF from DF. Can you clarify that here?*

We suggest to modify the original sentence

"In principle, it should be enough to to determine the time-independent coefficients from a single active interval."

by adding the clarification at the end of it:

"In principle, it should be enough to to determine the time-independent coefficients from a single active interval modelled using SMAP and PGIEM2G."

*Line 347: good data are available*

Will be corrected.

*Figure 3 caption: Last sentence says Bx, By, Bz but that is -B_theta, B_phi, -B_r rather than r, theta, phi as written.*

Will be corrected.

*Figure 5: Conductivity is in a diverging blue-white-red color scale - could you change it to a linear one (i.e. no white in the middle). This applies to other figures or plots with linear increasing rather than positive/negative variations*

If it is acceptable, we would like to keep the current color scale. The diverging scale makes it easy to separate small and large values, which is important for the electric field amplitude and conductances. Furthermore, the combination of red and blue should be suitable for the colorblind, whereas many other color scales are not.

*Figure 10: similar comment about linearly increasing colors. Also there doesn't seem to be any red in the plots.*

There are very small areas of red, mainly at some coast lines. However, as this is not clear, we suggest to saturate the plots further, as shown in the attached Figure 1.

*Figure 18: the label on the colorbars are not legible*

We will make them larger as shown in the attached Figure 2.

In addition, we will remove some extra $\epsilon_0$ from Eq. 22 and Eq. 24. This is just a typo and does not affect the results.

[Figure]

Figure 1.

(a) Maximum $|\boldsymbol{E}|/|\boldsymbol{E}_{DF}|$

(b) Minimum $|\boldsymbol{E}|/|\boldsymbol{E}_{DF}|$

[Figure]

(c) $\boldsymbol{E}_{DF}$ direction for maximum $|\boldsymbol{E}|/|\boldsymbol{E}_{DF}|$

[Figure]

[Figure]

Figure 2.

---

## Referee Report (RR1)

**Title**: **Estimation of the 3-D geoelectric field at the Earth's surface using spherical elementary current systems**

**Manuscript ID: egusphere-2024-2831**

**Recommendation: Minor Revision**

**1 General Comments**

This is a very interesting paper which presents comparisons of many different methods for estimating geoelectric fields, including a novel SECS-based method. I have no concerns about the scientific content of the paper and thus my suggestions are primarily meant to improve readability for the intended audience.

The paper as currently written presents the different methods in a confusing way and often introduces new methods and ideas midway through the paper. As a result, the paper feels scattered and disjointed and I think several sections could be removed. To me, it would be much more logical to have a structure like this:

> Introduction: "In this paper, we are going to develop a new method for modelling geoelectric fields using SECS, and then we are going to compare that method to other existing methods such as Method A, B, C."

> Methods: Develop new SECS method and summarize all other methods in short sub-sections.

> Results: Show comparisons of novel method with Methods A, B, C in coherent figures.

> Discussion: Based on the results, modify SECS to include SECS+CF correction and decide that SECS+CF is preferred method. Discuss SECS+CF in relation to dH/dt, geoelectric polarization, and ionospheric fields.

*Figures*

It is difficult to make visual comparisons to the different modelling methods investigated. Here are some suggestions to make this section more readable:

(1) Only include one epoch in the main text. There is nothing in the text that requires showing more than one epoch. For completeness, you could place the second epoch in a supplementary material.

(2) Re-arrange figures and panels in a more logical fashion. The way the figures are arranged makes the comparison very difficult. Consider this sentence: "*The difference can be seen by comparing the respective DF electric fields in the bottom right panel of Fig. 3 and top right panel of Fig. 7 or the bottom right panel of Fig. 4 and top right panel of Fig. 8.*" This is just unnecessarily confusing for the reader to keep track of the figures on different mismatched panels for multiple epochs.

(3) Even if you rearrange the figures, it is still often difficult to visually see the differences between the methods. Consider including additional figures which subtract one method from the other.

**Minor Comments**

Lines 29-53: This section should be moved to the SECS theory section, with the remaining introduction modified appropriately.

Line 145: What is $R'$ variable?

Line 187: The $\sigma$ value here is not a constant value but varies with position. Is there some way that you can denote this location dependence (e.g. $\sigma(\theta, \phi, r)$?

Line 197: The units of $Q_{CF}$ are Volts? You provide an equation for the DF SECS (Equation 9), but do not provide an equation for the CF SECS. At the very least, a reference here would be helpful.

Line 203: This is the first time you have referenced grid cells. The entire Section 2 is only relevant for a single SECS pole, yes? But in reality, you are constructing model from a superposition of multiple poles. This should be clarified somewhere in Section 2.

Figure 3: How much are the internal components influenced by the local geology around a single magnetometer? For example, the ICK station has a very strong positive internal dB/dt at 23:16 which appears like a bullseye around that location. This corresponds to a very strong negative internal $E_{DF}$. The total $E_{DF}$ seems to rotate around this single station. Could this apparent structure be due to a relatively small-scale geological feature local to ICK?

Figure 5: Is this the integrated conductance of the upper 0-10 km? Or is the first layer of the SMAP model 10 km thick? For clarity, I would recommend plotting several slices of the model which includes the true surface layer (i.e. the Gulf of Bothnia should have conductivity near that of sea water), as well as some additional intermediary slices at e.g. 0.5 km, 1 km, 5 km, etc.

Line 243: The conductance should be the sum of the conductivity in each vertical layer multiplied by the thickness of the layer. The way you have written Equation 25 does not make sense unless the first layer of the SMAP model is 10 km thick.

Figure 6: Are you interpolating the 0.5° x 1° DF SECS poles onto the 5' x 5' CF SECS poles grid prior to adding them together to get the total field?

Line 263: The total electric field is generally higher in areas of lower conductivity and lower in areas of higher conductivity because of the inductive (DF internal) component of the electric field regardless of whether there are any nearby conductivity gradients.

Line 274: Is the top of the PGIEM2G model space set at the SECS altitude, or is there padding above the SECS altitude?

Line 280: The terminology here is confusing. You are using PGIEM2G to estimate SECS amplitudes and then comparing them to SECS amplitudes?

Line 294: "SECS method reconstructs it from ground-based geomagnetic measurements"

Line 315-324: I am a bit confused by this section because you are using the *total* (internal+external) DF electric field structure. Even if you satisfy the plane-wave external source, you could still have small-scale aberrations in the internal DF electric field due to variations in ground structure (see previous comment on Figure 3 at ICK).

Line 328-329: "original PGIEM2G-modelled"

Line 342: "SECS-modelling based on PGIEM2G magnetic field". Which method is this referring to?

Line 343-346: The MSTF method should be mentioned earlier and explained in more detail.

Line 350-351: Figure 15 shows the geoelectric field modelled with the original SECS only, or the SECS with the CF correction applied?

Line 355-360: What is causing the temporal variation in the DF and CF behavior? It is unintuitive to me that sometimes the internal DF would dominate while other times the external DF would dominate since the ratio of internal to external should be mostly determined by the fixed conductivity structure.

Section 3.5.1: This section should be removed. As far as I can tell, this section makes no reference to the new SECS method. You also specifically draw attention to the fact that your SECS and PGIEM2G geoelectric estimates are not possible: "the magnetometer network … was very sparse and the conductivity model in this region is known to be inaccurate".

Section 3.5.2: This section should be removed as well, for similar reasons as above. Once again, as far as I can tell, you aren't actually using any of the new methods to estimate the geoelectric field. You are just approximating the electric field using a simple relation to dB/dt at MEK (Equations 39, 40). It completely undermines the point of your paper to develop this SECS method and then say, "we can't use this new method, so we are going to use a simpler method instead".

Section 4.1: This section is confusing and unexpected. Based on the paper thus far (excluding Section 3.5), I thought the entire purpose of the paper was to compare different methods of geoelectric field estimation. So why would you not compare the different methods' peak values? To me, that is the fundamental quantity for space weather hazards: *how big is the largest estimated geoelectric field and when and where does it occur*? Do the different SECS, PGIEM2G, MSTF, or 1-D methods result in different peak total electric fields? Are they different by a large amount (e.g. an order of magnitude) or a very tiny amount (e.g. 2%)? Do the peaks occur at different times or different locations?

Section 4.1: It is a bit perplexing that you only discuss the time of the peak, without ever mentioning in the text that the peak total geoelectric field is *72 V/km*. Pulkkinen et al. (2012) cite 20 V/km as an extreme 1-in-100 year geoelectric field event, and Lanabere et al. (2024) cite a value of 8.5 V/km as extreme for Sweden.  Do you think that your >70 V/km value is realistic? Over what spatial and temporal scale does that peak value occur? More discussion of this is required.

Line 415-422: Continuing from the previous comment: from a space weather hazards perspective, it isn't very relevant when the DF or CF peaks occurred. The only thing that really matters is when the *total* peak occurs, and how different modelling methods compare in this regard. The DF and CF peaks are mathematical abstractions that don't actually occur in reality.

Line 425: The discussion about dH/dt is interesting and definitely relevant to the scientific community. You could improve this by computing the correlation coefficient between dH/dt versus E field. You could also specifically compare DF, CF and total (DF+CF) fields, with the expectation that the DF component would have the strongest correlation with dH/dt based on Equations 42 and 43.

Section 4.2: Line 435-440 could be mentioned elsewhere as a single sentence in the conclusion around Line 542. If you remove Line 435-440, then this section should be renamed "directionality of the E field" or something similar. Your discussion of polarization is very interesting since it represents a way of deriving E-field polarizations independent of standard MT techniques (e.g. polarization ellipses, Love et al., 2022). The importance of E-polarization in GIC studies has been highlighted in several papers (Cordell et al., 2021; Love et al., 2019, 2022; Malone-Leigh et al., 2024; Murphy et al., 2021). It would be interesting to explore this in more detail to discern if your polarizations match standard MT polarizations.

Section 4.3: This section should be removed, but I do think that a comparison to a 1-D model in the results section could be a useful addition since it is still common in the power industry. Showing how and why the 1-D method fails is useful.

Section 4.4: This is a very interesting section which could probably be it's own paper. Including the effects of Earth induction have been mostly ignored by the space physics community, especially those trying to run large MHD models to predict ground magnetic field behaviour (e.g. SWMF, MAGE, etc.).

Section 4.5: I am not sure if this needs to be its own section. You could probably just include a small statement about the permittivity when you introduce Equation 24.

**References**

Cordell, D., Unsworth, M. J., Lee, B., Hanneson, C., Milling, D. K., & Mann, I. R. (2021). Estimating the Geoelectric Field and Electric Power Transmission Line Voltage During a Geomagnetic Storm in Alberta, Canada Using Measured Magnetotelluric Impedance Data: The Influence of Three-Dimensional Electrical Structures in the Lithosphere. *Space Weather*, *19*(10), e2021SW002803. https://doi.org/10.1029/2021SW002803

Love, J. J., Lucas, G. M., Bedrosian, P. A., & Kelbert, A. (2019). Extreme-Value Geoelectric Amplitude and Polarization Across the Northeast United States. *Space Weather*, *17*(3), 379–395. https://doi.org/10.1029/2018SW002068

Love, J. J., Lucas, G. M., Rigler, E. J., Murphy, B. S., Kelbert, A., & Bedrosian, P. A. (2022). Mapping a Magnetic Superstorm: March 1989 Geoelectric Hazards and Impacts on United States Power Systems. *Space Weather*, *20*(5), e2021SW003030. https://doi.org/10.1029/2021SW003030

Malone-Leigh, J., Campanyà, J., Gallagher, P. T., Hodgson, J., & Hogg, C. (2024). Mapping Geoelectric Field Hazards in Ireland. *Space Weather*, *22*(2), e2023SW003638. https://doi.org/10.1029/2023SW003638

Murphy, B. S., Lucas, G. M., Love, J. J., Kelbert, A., Bedrosian, P. A., & Rigler, E. J. (2021). Magnetotelluric Sampling and Geoelectric Hazard Estimation: Are National-Scale Surveys Sufficient? *Space Weather*, *19*(7), e2020SW002693. https://doi.org/10.1029/2020SW002693

---

## Author Response (AR2)

**Replies to Referees #1 and #3**

Liisa Juusola[1], Heikki Vanhamäki[2], Elena Marshalko[1], Mikhail Kruglyakov[3], and Ari Viljanen[1]

[1]Finnish Meteorological Institute, Helsinki, Finland
[2]University of Oulu, Oulu, Finland
[3]University of Otago, Dunedin, New Zealand

**Dear Referee #1,**

thank you again for constructive comments. Please, see below for our point-by-point replies. The original review is written in *black* and our replies in blue. The line numbers refer to the revised manuscript.

*Review comments on the revised manuscript egusphere-2024-2831, entitled: Estimation of the 3-D geoelectric field at the Earth's surface using Spherical Elementary Current Systems*

*by Liisa Juusola et al.*

*The authors have well responded to all my comments. In this for I regard the manuscript almost ready for publication. There are just a few suggestions that might be worth to be considered before acceptance.*

1. *line 68: The sentence "Their approach ...." is very long and difficult to digest. It should be cut at least into two parts.*

   We have cut the sentence as suggested.

2. *Sect. 2: The title of the paper suggests a general relevance, e.g. for GIC and magnetotelluric studies. However, the SECS method introduced by Amm (1997) is based on vertical CF currents. This prerequisite is sufficiently fulfilled only at high latitudes where FACs are flowing almost vertical to the Earth's surface, but the approach loses applicability at middle and low latitudes. This limitation to high latitudes has to be mentioned already in the Introduction and possibly again at later stages.*

   Fortunately, this is not so. The ground magnetic field variations, and thus the divergence-free electric field, can be described everywhere on Earth's surface in terms of ionospheric and telluric divergence-free equivalent currents. However, the equivalent currents can only be interpreted in terms of the divergence-free part of the ionospheric horizontal current density at high latitudes where the FACs can be assumed to be radial. We have added a note on this at lines 157-160.

3. *line 529: I have the impression that you solve in Eq. (24) $Q_i$, not for $Q_i/\epsilon_o$. Please check it.*

   You are correct. We have removed this section as irrelevant.

4. *There are some hands full of small typos in the manuscript. They may best be found by a person who has not been involved in writing the manuscript. But this action could well be part of the proof reading.*

   That would be good. We have tried our best, but because English is not a native language for any of us, there are bound to be mistakes.

**Dear Referee #3,**

thank you for the careful reading of our manuscript and the constructive comments. Please, see below for our point-by-point replies. The original review is written in *black* and our replies in blue. The line numbers refer to the revised manuscript.

*Title: Estimation of the 3-D geoelectric field at the Earth's surface using spherical elementary current systems*

30 *Manuscript ID: egusphere-2024-2831*

*Recommendation: Minor Revision*

*1 General Comments*

*This is a very interesting paper which presents comparisons of many different methods for estimating geoelectric fields, including a novel SECS-based method. I have no concerns about the scientific content of the paper and thus my suggestions are*
35 *primarily meant to improve readability for the intended audience.*

*The paper as currently written presents the different methods in a confusing way and often introduces new methods and ideas midway through the paper. As a result, the paper feels scattered and disjointed and I think several sections could be removed. To me, it would be much more logical to have a structure like this:*

*Introduction: "In this paper, we are going to develop a new method for modelling geoelectric fields using SECS, and*
40 *then we are going to compare that method to other existing methods such as Method A, B, C."*

*Methods: Develop new SECS method and summarize all other methods in short sub-sections.*

*Results: Show comparisons of novel method with Methods A, B, C in coherent figures.*

*Discussion: Based on the results, modify SECS to include SECS+CF correction and decide that SECS+CF is preferred method. Discuss SECS+CF in relation to dH/dt, geoelectric polarization, and ionospheric fields.*

45 The paper has now been re-organized: We have rewritten parts of the Introduction to better explain the content of the paper, as suggested. Method development and summaries of existing methods have been moved to the Methods-section. We have not moved the discussion on selecting the preferred method to the Discussion, because we consider it to be an integral part of the Results section. Our logic is that the Results section contains the testing and validation of the new method and the Discussion section contains discussion on possible applications of the new method. Hopefully this is an acceptable approach. We have
50 tried to make this logic more clear by adding preliminary words for the Methods and Discussion sections.

*Figures*

*It is difficult to make visual comparisons to the different modelling methods investigated. Here are some suggestions to make this section more readable:*

*(1) Only include one epoch in the main text. There is nothing in the text that requires showing more than one epoch. For*
55 *completeness, you could place the second epoch in a supplementary material.*

Corrected as suggested.

*(2) Re-arrange figures and panels in a more logical fashion. The way the figures are arranged makes the comparison very difficult. Consider this sentence: "The difference can be seen by comparing the respective DF electric fields in the bottom right panel of Fig. 3 and top right panel of Fig. 7 or the bottom right panel of Fig. 4 and top right panel of Fig.*
60 *8." This is just unnecessarily confusing for the reader to keep track of the figures on different mismatched panels for multiple epochs.*

We have tried to do this as much as possible, but because it is not possible to fit everything into the same Figure, we have also added labels to the panels to avoid confusing sentences such as the one cited above.

*(3) Even if you rearrange the figures, it is still often difficult to visually see the differences between the methods. Consider including additional figures which subtract one method from the other.*

Due to the differences in grids and analysis areas, this would not be straightforward to implement. Fig. 10 and A4 already have such panels.

*Minor Comments*

*Lines 29–53: This section should be moved to the SECS theory section, with the remaining introduction modified appropriately.*

Corrected as suggested.

*Line 145: What is $R'$ variable?*

The variable is $R$, but a comma in the sentence made it look like $R'$. We have removed the comma.

*Line 187: The $\sigma$ value here is not a constant value but varies with position. Is there some way that you can denote this location dependence (e.g. $\sigma(\theta,\phi,r)$?*

Corrected as suggested.

*Line 197: The units of $Q_{CF}$ are Volts? You provide an equation for the DF SECS (Equation 9), but do not provide an equation for the CF SECS. At the very least, a reference here would be helpful.*

Yes, the units are Volts, because $Q_{CF}$ is charge divided by $\epsilon_0$. We have added a note about this. We have also added a reference for the CF SECS equation.

*Line 203: This is the first time you have referenced grid cells. The entire Section 2 is only relevant for a single SECS pole, yes? But in reality, you are constructing model from a superposition of multiple poles. This should be clarified somewhere in Section 2.*

We have added: "The equations above apply to a single SECS pole." at line 238.

*Figure 3: How much are the internal components influenced by the local geology around a single magnetometer? For example, the ICK station has a very strong positive internal dB/dt at 23:16 which appears like a bullseye around that location. This corresponds to a very strong negative internal $\mathbf{E}_{DF}$. The total $\mathbf{E}$ seems to rotate around this single station. Could this apparent structure be due to a relatively small-scale geological feature local to ICK?*

It is true that the telluric equivalent currents and associated $\partial\mathbf{B}/\partial t$ and $\mathbf{E}_{DF}$ tend to show smaller scale sizes than the ionospheric equivalent currents. The internal feature around JCK (Fig. 3b and e) is not that different from the other areas. What makes it stand out in the total field (Fig. 3c and f) is that, unlike the other areas, $dB_z/dt$ and the curl have the same sign as the external fields (Fig. 3a and d). At 23:52 (Fig. A1), JCK shows a similar behavior as at 23:16, but this time as a part of a larger feature that encompasses LYC as well. The conductivity map (Fig. 4) shows that JCK is located in an area where there are sharp conductivity gradients.

*Figure 5: Is this the integrated conductance of the upper 0-10 km? Or is the first layer of the SMAP model 10 km thick? For clarity, I would recommend plotting several slices of the model which includes the true surface layer (i.e. the Gulf of Bothnia should have conductivity near that of sea water), as well as some additional intermediary slices at e.g. 0.5 km, 1 km, 5 km, etc.*

The first layer of SMAP is 10 km thick. We have tried to clarify this in the text.

*Line 243: The conductance should be the sum of the conductivity in each vertical layer multiplied by the thickness of the layer. The way you have written Equation 25 does not make sense unless the first layer of the SMAP model is 10* km *thick.*

That is true, but the first layer is indeed 10 km thick.

*Figure 6: Are you interpolating the $0.5^o \times 1^o$ DF SECS poles onto the $5' \times 5'$ CF SECS poles grid prior to adding them together to get the total field?*

No, we use the $0.5^o \times 1^o$ grid for the DF SECS poles but calculate the DF electric field on the $5' \times 5'$ grid. The resolution of the DF SECS pole grid is determined by the density of the magnetometers, no benefit would be gained by using a denser grid. Of course, the output on the denser grid cannot contain scale sizes smaller than the DF SECS pole grid (cf. line 407).

*Line 263: The total electric field is generally higher in areas of lower conductivity and lower in areas of higher conductivity because of the inductive (DF internal) component of the electric field regardless of whether there are any nearby conductivity gradients.*

The sentence has been reformulated: "As expected, charges accumulate at conductivity gradients, producing a total electric field that is significantly modulated compared to the DF electric field of Fig. 3f. Especially, the tendency for higher electric field in areas of lower conductivity and lower electric field in areas of higher conductivity is more pronounced."

*Line 274: Is the top of the PGIEM2G model space set at the SECS altitude, or is there padding above the SECS altitude?*

As the integral equation approach is invoked, the model space is infinite. This has been clarified at lines 290.

*Line 280: The terminology here is confusing. You are using PGIEM2G to estimate SECS amplitudes and then comparing them to SECS amplitudes?*

The text has been reformulated: "The 2-D SECS method can also be used to separate any vector field on a sphere into its DF and CF parts (Amm, 1997). We have utilized it to separate the total electric field produced by PGIEM2G. The DF part of the electric field..."

*Line 294: "SECS method reconstructs it from ground-based geomagnetic measurements"*

Corrected as suggested.

*Line 315–324: I am a bit confused by this section because you are using the total (internal+external) DF electric field structure. Even if you satisfy the plane-wave external source, you could still have small-scale aberrations in the internal DF electric field due to variations in ground structure (see previous comment on Figure 3 at ICK).*

This is true, but as can be seen when comparing the internal DF electric field in Fig. 3e and the CF electric field in Fig. 5e and 6c, the CF electric field features tend to be smaller than the internal DF electric field features. We do not require that the external or total field needs to be a plane wave, only that, locally, the DF electric field structures are larger than the range of the relevant charge accumulations that contribute to the local CF field. The validity of this assumption is illustrated in Fig. 9, 10, and A4.

*Line 328–329: "original PGIEM2G-modelled"*

Corrected as suggested.

*Line 342: "SECS-modelling based on PGIEM2G magnetic field". Which method is this referring to?*

The sentence has been reformulated: "PGIEM2G modelling is shown in black, DF SECS modelling based on IMAGE magnetic field with the CF part obtained using the coefficients from Fig. 7 in red, and DF SECS modelling based on PGIEM2G magnetic field with the CF part obtained using the coefficients from Fig. 7 in green."

*Line 343–346: The MSTF method should be mentioned earlier and explained in more detail.*

The MSTF method is now described in Section 2.5 in more detail.

*Line 350–351: Figure 15 shows the geoelectric field modelled with the original SECS only, or the SECS with the CF correction applied?*

The text has been reformulated: "Finally, Figure 12 shows a time series of the electric field as a sum of the external DF, internal DF, and CF parts, at the location of the magnetometer station SOD between 7 September 2017 23:00 UT and 8 September 2017 01:00 UT. The DF electric field has been modelled using the DF SECS method and IMAGE data, and the CF electric field has been obtained using the coefficients from Fig. 7."

*Line 355–360: What is causing the temporal variation in the DF and CF behavior? It is unintuitive to me that sometimes the internal DF would dominate while other times the external DF would dominate since the ratio of internal to external should be mostly determined by the fixed conductivity structure.*

This is a very good question! We have added discussion on this at lines 503–510: "The temporal variation in the DF and CF electric field behavior in Fig. 12 is caused by the combination of the spatiotemporally complex external source and the dynamic response of the fixed 3-D ground conductivity. Although induction in the 3-D ground is deterministic, it is unpredictable and difficult to understand, as mentioned in the Introduction. Ideally, we could determine the electric field everywhere in the earth, calculate the telluric current density and charge density, and visualize them in 3-D to understand the complete induction process. However, this is not feasible with present computational resources. Our new method can be used to analyse the spatiotemporal variations of the different geoelectric field contributions: time-variations of ionospheric currents, time-variations of telluric currents, and accumulated charges in the ground. This can help in clarifying the complicated interaction between the ionosphere and the conducting ground."

*Section 3.5.1: This section should be removed. As far as I can tell, this section makes no reference to the new SECS method. You also specifically draw attention to the fact that your SECS and PGIEM2G geoelectric estimates are not possible: "the magnetometer network ... was very sparse and the conductivity model in this region is known to be inaccurate".*

It is true that we are not using the SECS method in this section. However, the simple relation used here and in section 3.5.2 is based on the same principle as the new method (cf. section 2.4). A key difference compared to the conventional Eq. 45 and 46 is the inclusion of $\partial B_r/\partial t$, which was derived using the principle of the DF SECS method.

*Section 3.5.2: This section should be removed as well, for similar reasons as above. Once again, as far as I can tell, you aren't actually using any of the new methods to estimate the geoelectric field. You are just approximating the electric field using a simple relation to dB/dt at MEK (Equations 39, 40). It completely undermines the point of your paper to develop this SECS method and then say, "we can't use this new method, so we are going to use a simpler method instead".*

The modified Introduction and the preliminary words to Section 2 are intended to better motivate the inclusion of this method. We believe that it is not an undermining but an extension of the SECS method.

*Section 4.1: This section is confusing and unexpected. Based on the paper thus far (excluding Section 3.5), I thought the entire purpose of the paper was to compare different methods of geoelectric field estimation. So why would you not compare the different methods' peak values? To me, that is the fundamental quantity for space weather hazards: how big is the largest estimated geoelectric field and when and where does it occur? Do the different SECS, PGIEM2G, MSTF, or 1-D methods result in different peak total electric fields? Are they different by a large amount (e.g. an order of magnitude) or a very tiny amount (e.g. 2%)? Do the peaks occur at different times or different locations?*

The logic of the text is that validation of the new method, including comparison with other methods, is carried out in Section 3, and further applications of the new method are discussed in Section 4. Thus, the idea behind Section 4.1 is not to compare different methods, but to demonstrate how the new method could be used to understand the formation of geoelectric field peaks. We have added text to explanain this at lines 497–501.

The questions you pose are relevant indeed, and should be considered in detail in a separate study.

*Section 4.1: It is a bit perplexing that you only discuss the time of the peak, without ever mentioning in the text that the peak total geoelectric field is 72 V/km. Pulkkinen et al. (2012) cite 20 V/km as an extreme 1-in-100 year geoelectric field event, and Lanabere et al. (2024) cite a value of 8.5 V/km as extreme for Sweden. Do you think that your >70 V/km value is realistic? Over what spatial and temporal scale does that peak value occur? More discussion of this is required.*

Thank you for the suggestion, we have added such discussion at lines 522–530.

*Line 415-422: Continuing from the previous comment: from a space weather hazards perspective, it isn't very relevant when the DF or CF peaks occurred. The only thing that really matters is when the total peak occurs, and how different modelling methods compare in this regard. The DF and CF peaks are mathematical abstractions that don't actually occur in reality.*

This is true, but separation into CF and DF part may provide a better understanding of the characteristics of the total field. Also, there is a fundamental physical basis, as explained in the Introduction: The CF and DF electric fields are related to charges and currents, respectively.

*Line 425: The discussion about dH/dt is interesting and definitely relevant to the scientific community. You could improve this by computing the correlation coefficient between dH/dt versus E field. You could also specifically compare DF, CF and total (DF+CF) fields, with the expectation that the DF component would have the strongest correlation with dH/dt based on Equations 42 and 43.*

This is an interesting suggestion, which could be carried out in a separate study.

*Section 4.2: Line 435-440 could be mentioned elsewhere as a single sentence in the conclusion around Line 542. If you remove Line 435-440, then this section should be renamed "directionality of the E field" or something similar. Your discussion of polarization is very interesting since it represents a way of deriving E-field polarizations independent of standard MT techniques (e.g. polarization ellipses, Love et al., 2022). The importance of E-polarization in GIC studies has been highlighted in several papers (Cordell et al.,2021; Love et al., 2019, 2022; Malone-Leigh et al., 2024; Murphy et al., 2021). It would be interesting to explore this in more detail to discern if your polarizations match standard MT polarizations.*

Corrected as suggested. We have also added discussion about polarization, including the references kindly provided.

*Section 4.3: This section should be removed, but I do think that a comparison to a 1-D model in the results section could be a useful addition since it is still common in the power industry. Showing how and why the 1-D method fails is useful.*

Corrected as suggested. We are currently preparing a separate study on a detailed comparison of 3-D and 1-D models.

*Section 4.4: This is a very interesting section which could probably be it's own paper. Including the effects of Earth induction have been mostly ignored by the space physics community, especially those trying to run large MHD models to predict ground magnetic field behaviour (e.g. SWMF, MAGE, etc.).*

Thank you for the suggestion, we have added a mention about this at the end of the section.

*Section 4.5: I am not sure if this needs to be its own section. You could probably just include a small statement about the permittivity when you introduce Equation 24.*

This section has been removed as irrelevant.

*References*

*Cordell, D., Unsworth, M. J., Lee, B., Hanneson, C., Milling, D. K., & Mann, I. R. (2021). Estimating the Geoelectric Field and Electric Power Transmission Line Voltage During a Geomagnetic Storm in Alberta, Canada Using Measured Magnetotelluric Impedance Data: The Influence of Three-Dimensional Electrical Structures in the Lithosphere. Space Weather, 19(10), e2021SW002803. https://doi.org/10.1029/2021SW002803*

Love, J. J., Lucas, G. M., Bedrosian, P. A., & Kelbert, A. (2019). Extreme-Value Geoelectric Amplitude and Polarization Across the Northeast United States. Space Weather, 17(3), 379–395. https://doi.org/10.1029/2018SW002068

225    Love, J. J., Lucas, G. M., Rigler, E. J., Murphy, B. S., Kelbert, A., & Bedrosian, P. A. (2022). Mapping a Magnetic Superstorm: March 1989 Geoelectric Hazards and Impacts on United States Power Systems. Space Weather, 20(5), e2021SW003030. https://doi.org/10.1029/2021SW003030

Malone-Leigh, J., Campanyà, J., Gallagher, P. T., Hodgson, J., & Hogg, C. (2024). Mapping Geoelectric Field Hazards in Ireland. Space Weather, 22(2), e2023SW003638. https://doi.org/10.1029/2023SW003638

230    Murphy, B. S., Lucas, G. M., Love, J. J., Kelbert, A., Bedrosian, P. A., & Rigler, E. J. (2021). Magnetotelluric Sampling and Geoelectric Hazard Estimation: Are National-Scale Surveys Sufficient? Space Weather, 19(7), e2020SW002693. https://doi.org/10.1029/2020SW002693